# Rethinking Invariance Regularization in Adversarial Training to Improve Robustness-Accuracy Trade-off

**Futa Waseda**[1]    **Ching-Chun Chang**[2]    **Isao Echizen**[1,2]
[1]The University of Tokyo    [2]National Institute of Informatics
`futa-waseda@g.ecc.u-tokyo.ac.jp, {ccchang,iechizen}@nii.ac.jp`

## Abstract

Adversarial training often suffers from a robustness-accuracy trade-off, where achieving high robustness comes at the cost of accuracy. One approach to mitigate this trade-off is leveraging invariance regularization, which encourages model invariance under adversarial perturbations; however, it still leads to accuracy loss. In this work, we closely analyze the challenges of using invariance regularization in adversarial training and understand how to address them. Our analysis identifies two key issues: (1) a "gradient conflict" between invariance and classification objectives, leading to suboptimal convergence, and (2) the mixture distribution problem arising from diverged distributions between clean and adversarial inputs. To address these issues, we propose **A**symmetric **R**epresentation-regularized **A**dversarial **T**raining (**AR-AT**), which incorporates asymmetric invariance loss with stop-gradient operation and a predictor to avoid gradient conflict, and a split-BatchNorm (BN) structure to resolve the mixture distribution problem. Our detailed analysis demonstrates that each component effectively addresses the identified issues, offering novel insights into adversarial defense. AR-AT shows superiority over existing methods across various settings. Finally, we discuss the implications of our findings to knowledge distillation-based defenses, providing a new perspective on their relative successes.

## 1 Introduction

Computer vision models based on deep neural networks (DNNs) are vulnerable to adversarial examples (AEs) (Szegedy et al., 2014; Goodfellow et al., 2015), which are carefully perturbed inputs to fool DNNs. Since AEs can fool DNNs without affecting human perception, they pose potential threat to real-world DNN applications. Adversarial training (AT) (Goodfellow et al., 2015; Madry et al., 2018) has been the state-of-the-art defense strategy, however, AT-based methods suffer from a significant robustness-accuracy trade-off (Tsipras et al., 2019): to achieve high robustness, they sacrifice accuracy on clean images. Despite the extensive number of studies, this trade-off has been a huge obstacle to their practical use.

One of the promising approaches to mitigate this trade-off is to leverage invariance regularization, such as TRADES (Zhang et al., 2019) and LBGAT (Cui et al., 2021), which encourage the model to be invariant under adversarial perturbations. However, these methods still face trade-offs, and their limitations require further understanding.

To this end, we investigate the following research question: *"How can a model learn adversarially invariant representations without compromising discriminative ability?."* This work carefully analyzes the challenges of applying invariance regularization in AT to improve the robustness-accuracy trade-off. We identify novel issues and propose novel solutions to address them, offering novel insights into adversarial defense.

Our investigation identifies two key issues in applying invariance regularization (Fig.1a): (1) a "gradient conflict" between invariance loss and classification objectives, leading to suboptimal convergence, and (2) the mixture distribution problem within Batch Normalization (BN) layers when the same BNs are used for both clean and adversarial inputs. "Gradient conflict" suggests that minimizing invariance loss may push the model toward non-discriminative directions and conflict with the classification

objectives. Furthermore, we find that using the same BNs for both clean and adversarial inputs causes the mixture distribution problem: the BNs are updated to have a mixture distribution of clean and adversarial inputs, which can be suboptimal for both clean and adversarial inputs.

To address these issues, we propose a novel method, **A**symmetric **R**epresentation-regularized **A**dversarial **T**raining (**AR-AT**), incorporating an asymmetric invariance regularization with a stop-gradient operation and a predictor, and a split-BN structure, as depicted in Fig. 1f. Our step-by-step analysis demonstrates that each component effectively addresses the identified issues, offering novel insights: stop-gradient operation resolves "gradient conflict," and split-BN structure resolves the mixture distribution problem. With both components combined, AR-AT improves the robustness-accuracy trade-off, outperforming existing methods across various settings.

Furthermore, we discuss the relevance of our findings to existing knowledge distillation (KD)-based defenses (Cui et al., 2021; Suzuki et al., 2023), pointing out the underlying factors of their effectiveness that have not yet been investigated. We attribute their effectiveness to resolving "gradient conflict" and the mixture distribution problem.

Our contributions are summarized as follows:

- We focus on understanding the challenges of invariance regularization in adversarial training to improve the robustness-accuracy trade-off. We identify novel issues and propose novel solutions.

- We reveal two key issues in using invariance regularization: (1) a "gradient conflict" between invariance loss and classification objectives, and (2) the mixture distribution problem within Batch Norm (BN) layers.

- We propose a novel method, AR-AT, which incorporates an asymmetric invariance regularization and a split-BN structure. Our detailed analysis shows that each component effectively adresses the identified issues, offering novel insights into adversarial defense.

- AR-AT outperforms existing methods across various settings.

- We present a new perspective on KD-based defenses, which have not been well understood.

## 2 PRELIMINARIES

### 2.1 ADVERSARIAL ATTACK

Let $x \in \mathbb{R}^d$ be an input image and $y \in \{1, \ldots, K\}$ be a class label from a data distribution $\mathcal{D}$. Let $f_\theta : \mathbb{R}^d \to \mathbb{R}^K$ be a DNN model parameterized by $\theta$. Adversarial attacks aim to find a perturbation $\delta$ that fools the model $f_\theta$ by solving the following optimization problem:

$$x' = x + \delta, \quad \text{where} \quad \delta = \arg\max_{\delta \in \mathcal{S}} \mathcal{L}(f_\theta(x + \delta), y) \tag{1}$$

where $x'$ is an adversarial example, $\mathcal{S}$ is a set of allowed perturbations, and $\mathcal{L}$ is a loss function. In this paper, we define the set of allowed perturbations $\mathcal{S}$ with $L_\infty$-norm as $\mathcal{S} = \{\delta \in \mathbb{R}^d \mid \|\delta\|_\infty \leq \epsilon\}$, where $\epsilon$ represents the size of the perturbations. The optimization of Eq.1 is often solved iteratively based on the projected gradient descent (PGD) (Madry et al., 2018).

### 2.2 ADVERSARIAL TRAINING

Adversarial training (AT), which augments the training data with AEs, has been the state-of-the-art approach to defend against adversarial attacks. Originally, Goodfellow et al. (2015) proposed to train the model with AEs generated by the fast gradient sign method (FGSM); in contrast, Madry et al. (2018) proposed to train a model with much stronger AEs generated by PGD, which is an iterative version of FGSM. Formally, the standard AT (Madry et al., 2018) solves the following optimization:

$$\min_\theta \mathbb{E}_{(x,y)\sim\mathcal{D}} \left[ \max_{\delta \in \mathcal{S}} \mathcal{L}(f_\theta(x + \delta), y) \right] \tag{2}$$

where the inner maximization problem is solved iteratively based on PGD.

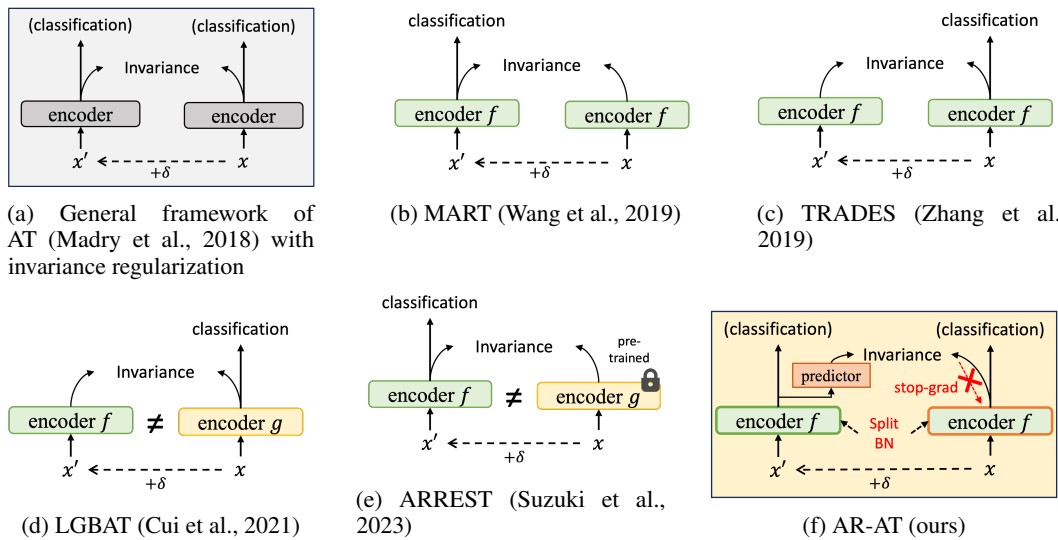

Figure 1: Comparison of invariance regularization-based adversarial defense methods. Our approach employs an asymmetric structure for invariance regularization with a stop-gradient and predictor, and a split-BatchNorm (BN) to maintain consistent batch statistics during training.

### 2.3 INVARIANCE REGULARIZATION-BASED DEFENSE

While AT only inputs AEs during training, invariance regularization-based adversarial defense methods (Zhang et al., 2019; Wang et al., 2019) input both clean and adversarial images to ensure adversarial invariance of the model. TRADES (Zhang et al., 2019) introduces a regularization term on the logits to encourage adversarially invariant predictions; although it allows trade-off adjustment by altering the regularization strength, it still suffers from a trade-off. MART (Wang et al., 2019) further improved TRADES by focusing more on the misclassified examples to enhance robustness, still sacrificing clean accuracy. Unlike these logit-based invariance regularization methods, our method leverages representation invariance to mitigate the trade-off as a generalized form of invariance regularization.

Another line of research is to employ knowledge distillation (KD) (Hinton et al., 2014)-based regularization, which has been shown effective in mitigating the trade-off. Cui et al. (2021) proposed LBGAT, which aligns the student model's predictions on adversarial images with the standard model's predictions on clean images, encouraging similarity of predictions between a student and a standardly trained teacher network. More recently, Suzuki et al. (2023) proposed ARREST, which performs representation-based KD so that the student model's representations are similar to the standardly trained model's representations. In contrast to these KD-based methods, which enforce adversarial invariance implicitly, we investigate output invariance within a single model, potentially more effective and more memory-efficient during training. Furthermore, we provide a new perspective on the relative success of KD-based methods, which had not been well understood.

Figure. 1 summarizes the loss functions of these methods and compares them with our method.

## 3 ASYMMETRIC REPRESENTATION-REGULARIZED ADVERSARIAL TRAINING (AR-AT)

In this section, we introduce a novel approach to effectively learn adversarially invariant representation without sacrificing discriminative ability on clean images.

A straightforward approach to learn adversarially invariant representation is to employ a siamese structured invariance regularization, depicted in Fig. 1a, as follows:

$$\mathcal{L}_{V0} = \alpha \cdot \mathcal{L}(f_\theta(x'), y) + \beta \cdot \mathcal{L}(f_\theta(x), y) + \gamma \cdot Dist(z, z') \tag{3}$$

where $x'$ is an AE generated from $x$ using Cross-Entropy Loss, $z$ and $z'$ are the normalized latent representations of $x$ and $x'$, respectively. $\mathcal{L}$ is a classification loss and $Dist$ is a distance metric. Although existing approaches typically rely on either adversarial or clean classification loss, we begin with this naive method to systematically identify and address the underlying issues step by step. Here, we focus on representation-based regularization since we found that it can be more effective in mitigating the trade-off than logit-based regularization, discussed in Appendix D.

However, we identify two key issues in this naive approach: (1) a "gradient conflict" between invariance loss and classification objectives, and (2) the mixture distribution problem arising from diverged distributions of clean and adversarial inputs. To address these issues, we propose a novel approach incorporating (1) a stop-gradient operation and a predictor MLP to asymmetrize the invariance regularization, and (2) a split-BatchNorm (BN) structure, explained in the following subsections.

## 3.1 ASYMMETRIZATION VIA STOP-GRADIENT FOR ADDRESSING GRADIENT CONFLICT

Although naive invariance regularization (Eq.3) is a straightforward approach for learning adversarially invariant representation, we observed a "gradient conflict" between invariance loss and classification objectives, as shown in Fig. 2. "Gradient conflict" occurs when the gradients of multiple loss functions oppose each other during joint optimization (i.e., for loss functions $L_A$ and $L_B$, $\nabla_\theta L_A \cdot \nabla_\theta L_B < 0$). Resolving this issue has been shown to enhance performance in various fields, including multi-task learning (Yu et al., 2020) and domain generalization (Mansilla et al., 2021). The observed "gradient conflict" suggests that minimizing the invariance loss may push the model toward non-discriminative directions, conflicting with the classification objectives, thereby degrading the classification performance.

In this work, we reveal that asymmetrizing the invariance regularization with a stop-gradient operation can effectively resolve "gradient conflict." The asymmetric invariance regularization is defined as follows:

$$\mathcal{L}_{V1} = \alpha \cdot \mathcal{L}(f_\theta(x'), y) + \beta \cdot \mathcal{L}(f_\theta(x), y) + \gamma \cdot Dist(z', \text{sg}(z)) \tag{4}$$

where $\text{sg}(\cdot)$ stops the gradient backpropagation from $z'$ to $z$, treating $z$ as a constant.

Our motivation of employing the stop-gradient operation is to eliminate the unnecessary gradient flow from the adversarial representation $z'$ to the clean representation $z$, which can lead to the degradation of classification performance. This can be understood by decomposing the invariance loss into two components, as follows:

$$Dist(z', z) = \left( Dist(z', \text{sg}(z)) + Dist(\text{sg}(z'), z) \right) / 2 \tag{5}$$

Minimizing the first term, $Dist(z', \text{sg}(z))$, encourages to bring the corrupted adversarial representation $z'$ closer to the clean representation $z$ by treating $z$ as a constant, which can be interpreted as the "purification" of representations. In contrast, minimizing the second term, $Dist(\text{sg}(z'), z)$, attempts to bring the clean representation $z$ closer to the potentially corrupted adversarial representation $z'$, encouraging the "corruption" of representations. This can be harmful for learning discriminative representations. Therefore, we hypothesize that minimizing the second term can conflict to the classification objectives, leading to the "gradient conflict."

## 3.2 LATENT PROJECTION FOR ENHANCING TRAINING STABILITY

We employ a predictor MLP $h$ to the latent representations $z'$ to predict $z$, inspired by recent self-supervised learning approach (Chen & He, 2021), as following:

$$\mathcal{L}_{V2} = \alpha \cdot \mathcal{L}(f_\theta(x'), y) + \beta \cdot \mathcal{L}(f_\theta(x), y) + \gamma \cdot Dist(h(z'), \text{sg}(z)) \tag{6}$$

The predictor MLP $h$ is trained to predict the clean representation $z$ from the corrupted adversarial representation $z'$. Our intuition is that the adversarial representation $z'$ varies with each training iteration, as adversarial perturbations depend on both the model parameters and the randomness inherent in the attack process. Therefore, introducing an additional predictor head may help stabilize the training process by handling the variations of adversarial representation through updates of the predictor, preventing large fluctuations of the classifier's parameters caused by variations in $z'$. In other words, the predictor MLP $h$ functions as a "stabilizer", preventing the model from being overly distracted by variations in adversarial representations while achieving invariance, thus preventing compromise of classification performance.

### 3.3 SPLIT BATCH NORMALIZATION FOR RESOLVING MIXTURE DISTRIBUTION PROBLEM

We point out that using the same Batch Norm (BN) (Ioffe & Szegedy, 2015) layers for both clean and adversarial inputs can lead to difficulty in achieving high performance due to a "mixture distribution problem." BNs are popularly used for accelerating the training of DNNs by normalizing the activations of the previous layer and then adjusting them via a learnable linear layer. However, since clean and adversarial inputs exhibit diverged distributions, using the same BNs on a mixture of these inputs can be suboptimal for both inputs, leading to reduced performance.

To address this issue, we employ a split-BN structure, which uses separate BNs for clean and adversarial inputs during training, inspired by Xie & Yuille (2019); Xie et al. (2020). Specifically, Eq. 6 is rewritten as follows:

$$\mathcal{L}_{V3} = \alpha \cdot \mathcal{L}(f_\theta(x'), y) + \beta \cdot \mathcal{L}(f_\theta^{\text{auxBN}}(x), y) + \gamma \cdot Dist(h(z'_{(\theta)}), \text{sg}(z_{(\theta^{\text{auxBN}})})) \tag{7}$$

where $f_\theta^{\text{auxBN}}$ shares parameters with $f_\theta$ but employs auxiliary BNs specialized for clean images, $z_{(\theta^{\text{auxBN}})}$ and $z'_{(\theta)}$ are the latent representations of $x$ and $x'$ from $f_\theta^{\text{auxBN}}$ and $f_\theta$, respectively. In this way, the BNs in $f_\theta$ exclusively process adversarial inputs, while the BNs in $f_\theta^{\text{auxBN}}$ exclusively process clean inputs, avoiding the mixture distribution problem.

Importantly, the split-BN structure is exclusively applied during training: During inference, the model $f_\theta$ is equipped to classify both clean and adversarial inputs with the same BNs. Therefore, in contrast to MBN-AT (Xie & Yuille, 2019) and AdvProp (Xie et al., 2020) that require test-time oracle selection of clean and adversarial BNs for optimal robustness and accuracy, our approach is more practical.

### 3.4 AUTO-BLANCE FOR REDUCING HYPERPARAMETERS

Furthermore, we employ a dynamic adjustment rule for the hyperparameters $\alpha$ and $\beta$, which automatically controls the balance between adversarial and clean classification loss. Specifically, we adjust $\alpha$ and $\beta$ based on the training accuracy of the previous epoch, inspired by Xu et al. (2023):

$$\alpha_{(t)} = Acc_{(t-1)} = \frac{1}{N} \sum_{i=1}^{N} \mathbb{I}(\text{argmax}(f_{\theta_{t-1}}^{\text{auxBN}}(x)) = y), \quad \beta_{(t)} = 1 - Acc_{(t-1)} \tag{8}$$

where $\alpha_{(t)}$ and $\beta_{(t)}$ are the hyperparameters at the current epoch $t$, and $Acc_{(t-1)}$ is the clean accuracy of the model $f_\theta^{\text{auxBN}}$ at the previous epoch $(t-1)$. This adjustment automatically forces the model to focus more on adversarial images as its accuracy on clean images improves. We empirically demonstrate that this heuristic dynamic adjustment strategy works well in practice, successfully reducing the hyperparameter tuning cost. We provide an ablation study in Sec. 5.5

### 3.5 MULTI-LEVEL REGULARIZATION OF LATENT REPRESENTATIONS

Finally, our method is extended to regularize multiple levels of representations. Specifically, we employ multiple predictor MLPs $h_1, \ldots, h_L$ to enforce invariance across multiple levels of representations $z_1, \ldots, z_L$ as follows: $\frac{1}{L} \sum_{l=1}^{L} Dist(h_l(z'_{(\theta),l}), \text{sg}(z_{(\theta^{\text{auxBN}}),l}))$. We found that regularizing multiple layers in the later stage of a network is the most effective, as demonstrated in our ablation study in Sec. 5.5.

## 4 EXPERIMENTAL SETUP

**Models and datasets.** We evaluate our method on CIFAR-10, CIFAR-100 (Krizhevsky & Hinton, 2009), and Imagenette (Howard, 2019) datasets. We use the standard data augmentation techniques of random cropping with 4 pixels of padding and random horizontal flipping. We use the model architectures of ResNet-18 (He et al., 2016) and WideResNet-34-10 (WRN-34-10) (Zagoruyko & Komodakis, 2016), following the previous works (Madry et al., 2018; Zhang et al., 2019; Cui et al., 2021). Implementation details of baselines are provided in Appendix A.3.

**Evaluation.** We use 20-step PGD attack (PGD-20) and AutoAttack (AA) (Croce & Hein, 2020) for evaluation. The perturbation budget is set to $\epsilon = 8/255$ with $l_\infty$-norm (Appendix B.4 shows that

Table 1: **Effectiveness of Stop-grad and Split-BN** on naive invariance regularization ($L_{V0}$). Split-BN consistently improves the robustness-accuracy trade-off. Stop-grad improves this trade-off, particulary when Split-BN addresses the "mixture distribution problem."

| | | Stop-grad | Split-BN | Clean | AA | Grad-sim. |
|---|---|---|---|---|---|---|
| **CIFAR10** | ResNet-18 | | | 82.93 | 46.50 | 0.06 |
| | | ✓ | | 82.47 | 45.21 | 0.68 |
| | | | ✓ | 84.35 | 47.96 | 0.14 |
| | | ✓ | ✓ | **85.51** | **49.30** | 0.59 |
| | WRN-34-10 | | | 86.25 | 41.17 | 0.03 |
| | | ✓ | | 86.60 | 42.04 | 0.31 |
| | | | ✓ | 87.13 | 47.31 | 0.01 |
| | | ✓ | ✓ | **88.27** | **47.98** | 0.23 |
| **CIFAR100** | ResNet-18 | | | 60.23 | 19.66 | 0.00 |
| | | ✓ | | 61.55 | 20.15 | 0.58 |
| | | | ✓ | 62.41 | **22.45** | -0.03 |
| | | ✓ | ✓ | **67.04** | 22.25 | 0.40 |
| | WRN-34-10 | | | 61.70 | 20.72 | -0.01 |
| | | ✓ | | 61.82 | 19.76 | 0.38 |
| | | | ✓ | 62.72 | 24.32 | 0.00 |
| | | ✓ | ✓ | **67.87** | **24.46** | 0.22 |

AR-AT is also effective for $l_2$-bounded scenarios). The step size is set to $2/255$ for PGD-20. We compare our method with the standard AT, and existing regularization-based methods TRADES, MART, LBGAT, and ARREST. Comparisons with other methods are provided in Appendix B.1.

**Training details.** We use a 10-step PGD for adversarial training. We initialized the learning rate to 0.1, divided it by a factor of 10 at the 75th and 90th epochs, and trained for 100 epochs. We use the SGD optimizer with a momentum of 0.9 and a weight decay of 5e-4, with a batch size of 128. Cosine Distance is used as the distance metric for invariance regularization. The predictor MLP has two linear layers, with the hidden dimension set to 1/4 of the feature dimension, following SimSiam (Chen & He, 2021). The latent representations to be regularized are spatially average-pooled to obtain one-dimensional vectors. The regularization strength $\gamma$ is set to 30.0 for ResNet-18 and 100.0 for WRN-34-10. We regularize all ReLU outputs in *"layer4"* for ResNet-18, and *"layer3"* for WRN-34-10. Computational cost of AR-AT is smaller than TRADES and LBGAT (Appendix G).

## 5 EMPIRICAL STUDY

### 5.1 ANALYSIS AND UNDERSTANDING OF EACH COMPONENT OF AR-AT

In this section, we analyze and understand the effectiveness of each component of our method, focusing on the stop-gradient operation, the split-BN structure, and the predictor MLP. Based on the naive invariance loss (Eq. 3), we add each component step by step to identify the underlying issues and address them systematically. Here, for simplicity, we omit the auto-balance heuristic and the multi-level regularization of latent representations, fixing the regularized layer to the last layer of the network for this analysis.

**Stop-gradient operation resolves "gradient conflict."** In Fig. 2a, we plot the cosine similarity between the gradients of the classification loss (the sum of clean and adversarial classification loss) and the invariance loss with respect to $\theta$ during training. The proportion of parameters experiencing gradient conflict in Fig. 2b. We observe that with the naive invariance regularization ($\mathcal{L}_{V0}$; Eq. 3), the gradient similarity fluctuates near zero, and over 40% of parameters experience gradient conflict, suggesting suboptimal convergence of both classification and invariance loss. To verify our hypothesis that the second term of Eq. 5, $D(sg(z'), z)$, conflicts with the classification loss, we isolate and minimize either the first or second term. We observe that minimizing only $D(sg(z'), z)$ results in strong gradient conflict. In contrast, minimizing only $D(z', sg(z))$ ($\mathcal{L}_{V1}$; Eq. 4) alleviates gradient conflict. This demonstrates the effectiveness of eliminating unnecessary gradient flow from the adversarial to clean representations, preventing their corruption and mitigating "gradient conflict."

However, Tab. 1 shows that mitigating "gradient conflict" does not necessarily improve the robustness and accuracy. Only in the cases of CIFAR-10/WRN-34-10 and CIFAR-100/ResNet-18, using the stop-gradient operation alone improved the robustness-accuracy trade-off. This suggests that the stop-gradient operation by itself is insufficient, which we address next.

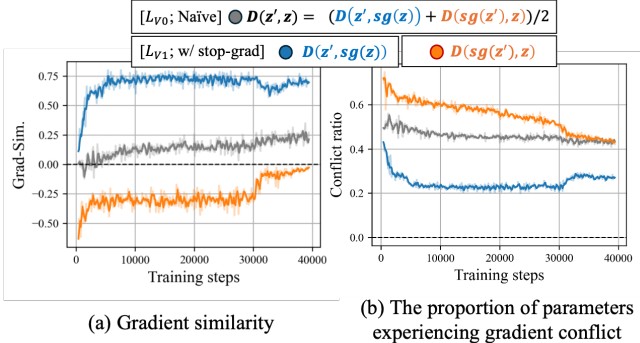

(a) Gradient similarity

(b) The proportion of parameters experiencing gradient conflict

Figure 2: **Gradient conflict between classification loss and different invariance losses**, w.r.t. $\theta$. With the naive invariance loss ($L_{V0}$), (a) gradient similarity is near zero, and (b) over 40% of parameters experience gradient conflicts. The term $D(sg(z'), z)$ causes strong conflict (orange line). In contrast, our approach ($L_{V1}$) to only minimize $D(z', sg(z))$ effectively resolves the conflict (blue line).

Figure 3: **Mixture distribution problem**, indicated by the L2 distance between adversarial and clean feature, $||z - z'||_2$. While $\mathcal{L}_{V0}$ already suffers from this problem, the use of stop-grad exacerbates this issue by weakening invariance regularization.

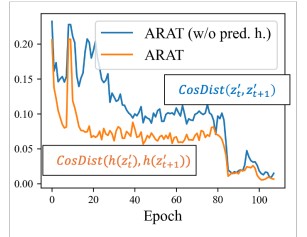

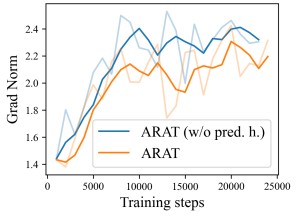

(a) Cosine distance between adversarial representations at epochs $t$ and $t + 1$.

(b) The gradient norm of the loss function ($||\nabla_\theta L||_2$).

Figure 4: **Predictor MLP head improves the training stability.** We observe that the predictor MLP stabilizes the updates of the representations where invariance loss is applied, and reduces the gradient norm.

Table 2: **Effectiveness of predictor MLP head** on the naive invariance regularization ($\mathcal{L}_{V0}$), using ResNet-18 (C10/100 refers to CIFAR-10/100).

|  | Stop-grad + Split-BN | Pred. h | Clean | AA |
|---|---|---|---|---|
| C10 | ✓ | | 85.51 | **49.30** |
| | ✓ | ✓ | **87.23** | **49.30** |
| C100 | ✓ | | 67.04 | 22.25 |
| | ✓ | ✓ | **67.64** | **22.42** |

**Split-BN resolves the mixture distribution problem.** Another issue in invariance regularization stems from using the same Batch Norm (BN) layers for both clean and adversarial inputs. Tab. 1 demonstrates that using split-BN structure consistently improves the robustness and accuracy. Importantly, Tab. 1 demonstrates that simultaneously resolving issues of "gradient conflict" and the mixture distribution problem leads to substantial improvements.

To understand the mixture distribution problem, we analyze the similarity between adversarial and clean features. In Fig. 3, we plot the L2 distance between the adversarial and clean features, $||z - z'||_2$, for the same input $x$. The increasing distance over time indicates that adversarial and clean features become more dissimilar during training, illustrating a robustness-accuracy trade-off. This suggests that shared BN layers between clean and adversarial inputs may fail to accurately estimate adversarial feature statistics (i.e., mixture distribution problem). Appendix F.2 demonstrates that the estimated batch statistics become more stable with the use of split-BN.

Notably, we observe that the stop-gradient operation exacerbates this issue by weakening the invariance regularization, as it removes the second term in Eq. 5. This explains why the stop-grad alone does not necessarily improve the robustness-accuracy trade-off, but does so when combined with split-BN, as shown in Tab. 1.

**Latent projection with predictor MLP head enhances performance by improving training stability.** Finally, we analyze the effectiveness of the predictor MLP head. Tab. 2 shows that employing the predictor MLP head ($\mathcal{L}_{V1}$ vs. $\mathcal{L}_{V2}$) leads to further improvements in the robustness

Table 3: **Comparison with invariance regularization-based defense methods on CIFAR datasets.** We report clean and robust accuracy (AutoAttack; AA). [1] [2]

| | Defense | CIFAR10 | | | CIFAR100 | | |
|---|---|---|---|---|---|---|---|
| | | Clean | AA | Sum. | Clean | AA | Sum. |
| ResNet-18 | AT | 83.77 | 42.42 | 126.19 | 55.82 | 19.50 | 75.32 |
| | TRADES | 81.25 | 48.54 | 129.79 | 54.74 | 23.60 | 78.34 |
| | MART | 82.15 | 47.83 | 129.98 | 54.54 | **26.04** | 80.58 |
| | LBGAT | $85.00_{\pm 0.47}$ | $48.85_{\pm 0.46}$ | $133.86_{\pm 0.65}$ | $65.87_{\pm 0.74}$ | $23.19_{\pm 0.74}$ | $89.07_{\pm 0.73}$ |
| | ARREST* | 86.63 | 46.14 | 132.77 | - | - | - |
| | **AR-AT (ours)** | $\mathbf{87.82}_{\pm 0.19}$ | $49.02_{\pm 0.47}$ | $\mathbf{136.84}_{\pm 0.33}$ | $\mathbf{67.51}_{\pm 0.13}$ | $23.38_{\pm 0.19}$ | $90.89_{\pm 0.29}$ |
| | **AR-AT+SWA (ours)** | $86.44_{\pm 0.05}$ | $\mathbf{50.28}_{\pm 0.14}$ | $136.72_{\pm 0.19}$ | $67.17_{\pm 0.18}$ | $24.36_{\pm 0.20}$ | $\mathbf{91.53}_{\pm 0.25}$ |
| WRN-34-10 | AT | 86.06 | 46.26 | 132.32 | 59.83 | 23.94 | 83.77 |
| | TRADES | 84.33 | 51.75 | 136.08 | 57.61 | 26.88 | 84.49 |
| | MART | 86.10 | 49.11 | 135.21 | 57.75 | 24.89 | 82.64 |
| | LBGAT | $88.19_{\pm 0.11}$ | $52.56_{\pm 0.34}$ | $140.75_{\pm 0.34}$ | $68.17_{\pm 0.56}$ | $26.92_{\pm 0.32}$ | $95.09_{\pm 0.64}$ |
| | ARREST* | 90.24 | 50.20 | 140.44 | **73.05** | 24.32 | 97.37 |
| | **AR-AT (ours)** | $\mathbf{90.89}_{\pm 0.22}$ | $50.77_{\pm 0.50}$ | $141.66_{\pm 0.51}$ | $72.51_{\pm 0.51}$ | $24.18_{\pm 0.46}$ | $96.70_{\pm 0.56}$ |
| | **AR-AT+SWA (ours)** | $90.06_{\pm 0.08}$ | $\mathbf{54.03}_{\pm 0.31}$ | $\mathbf{144.09}_{\pm 0.39}$ | $72.37_{\pm 0.13}$ | $\mathbf{27.31}_{\pm 0.13}$ | $\mathbf{99.69}_{\pm 0.26}$ |

Table 4: **Comparison with defense methods on Imagenette.** (The learning rate of LBGAT[†] is lowered from default due to gradient explosion.)

| | Defense | Imagenette | | |
|---|---|---|---|---|
| | | Clean | AA | Sum. |
| ResNet-18 | AT | 84.58 | 52.25 | 136.83 |
| | TRADES | 79.21 | 53.98 | 133.19 |
| | MART | 84.07 | **59.89** | 143.96 |
| | LBGAT[†] | 80.80 | 50.42 | 131.22 |
| | AR-AT (ours) | **88.66** | 59.28 | **147.94** |

Table 5: **Representation invariance analysis.** "Cos. sim.": cosine similarity between adversarial and clean features.

| | Defense | CIFAR10 | |
|---|---|---|---|
| | | Sum. | Cos. sim. |
| ResNet-18 | AT | 126.19 | 0.9423 |
| | TRADES | 129.79 | 0.9693 |
| | MART | 129.98 | 0.9390 |
| | LBGAT | 133.86 | 0.9236 |
| | AR-AT (ours) | 136.84 | 0.9450 |

and accuracy. In Fig. 4, we analyze the training stability. Fig. 4a shows the cosine distance between the adversarial representations at epochs $t$ and $t+1$, where the predictor MLP stabilizes the updates of the representations where invariance loss is applied. This suggests that the predictor MLP head helps stabilize the optimization of the invariance regularization by smoothing the updates of the representations. Fig. 4b reveals that, indeed, the predictor MLP reduces the gradient norm of the loss function ($||\nabla_\theta L||_2$), implying more stable optimization.

## 5.2 COMPARISON WITH STATE-OF-THE-ART INVARIANCE REGULARIZATOIN-BASED DEFENSES

In this section, we evaluate the complete version of AR-AT, which employs all components outlined in Sec. 3. The ablation study for the auto-balance heuristic and the multi-layer invariance regularization is provided in Sec. 5.5.

Tab. 3 shows the results of ResNet-18 and WRN-34-10 trained on CIFAR-10 and CIFAR-100, compared with existing invariance regularization-based defense methods. Following common practice in existing works (Sitawarin et al., 2021; Suzuki et al., 2023), we also report the sum of clean and robust accuracy against AA (i.e., *Sum.*) for the trade-off metric. We include the trade-off plots in Appendix B.2. We observe that our method achieves much better performance than the baselines of regularization-based defenses TRADES and MART on both datasets, highlighting the effectiveness of our methodology in employing invariance regularization. Moreover, our method achieves the best trade-off in most cases, outperforming LBGAT and ARREST, the state-of-the-art invariance regularization-based defenses. Additionally, we found that combining AR-AT with stochastic weight averaging (SWA) (Izmailov et al., 2018) enhances its performance, achieving the best results. Tab. 4 shows the results on Imagenette, a dataset with high-resolution images, and demonstrate that our method achieves the best performance. These results highlight that AR-AT not only introduces novel

---

[1]Results with * are directly copied from original papers.

[2]We report the mean and standard deviation of the results over five runs for LBGAT and AR-AT.

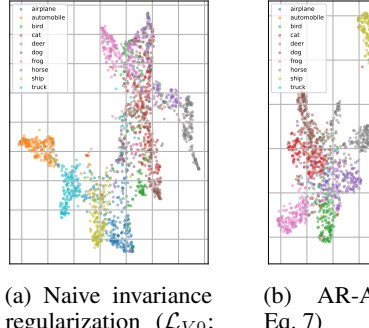

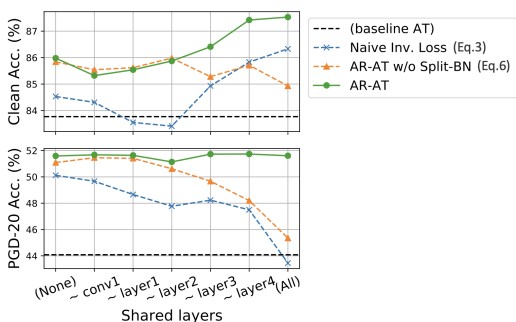

(a) Naive invariance regularization ($\mathcal{L}_{V0}$; Eq. 3)

(b) AR-AT ($\mathcal{L}_{V3}$; Eq. 7)

Figure 5: Visualization of clean representations for (a) naive invariance regularization ($\mathcal{L}_{V0}$, Eq. 3) and (b) AR-AT ($\mathcal{L}_{V3}$, Eq. 7) using UMAP (McInnes et al., 2018). 2000 images are randomly sampled from the CIFAR10 test set.

Figure 6: **Comparison of "sharing parameters vs. separate networks" in AR-AT for adversarial and clean branches.** The x-axis represents the depth until which parameters are shared: the leftmost corresponds to completely separate networks for adversarial and clean branches, while the rightmost shares all parameters (default).

perspectives to invariance regularization but also achieves state-of-the-art performance in mitigating the robustness-accuracy trade-off.

Note that the trade-off can be adjusted by simply changing the perturbation size $\epsilon$ in the adversarial training: In Appendix B.3, we show that AR-AT with $\epsilon = 9/255$ achieves both higher clean and robust accuracy than LBGAT at the same time.

### 5.3 ANALAYSIS OF LEARNED REPRESENTATIONS: ADVERSARIAL INVARIANCE AND DISCRIMINATIVE ABILITY

**Quantitative analysis.** Tab. 5 presents representation invariance measured by cosine similarity between adversarial and clean features extracted from the penultimate layer. While TRADES achieves high adversarial invariance, it sacrifices discriminative ability, as shown by its low accuracy. Conversely, LBGAT, a KD-based method, achieves high performance but low adversarial invariance, demonstrating that using a separate teacher network for regularization does not ensure invariance within a model. On the other hand, AR-AT achieves both high adversarial invariance and high accuracy simultaneously. We attribute this to effectively addressing the "gradient conflict" and the mixture distribution problem in invariance regularization, while imposing the invariance regularization directly on the model itself, which is different from KD-based methods.

**Visualization of learned representations.** Fig. 5 visualizes the clean representations for (a) naive invariance regularization ($\mathcal{L}_{V0}$, Eq. 3) and (b) AR-AT ($\mathcal{L}_{V3}$, Eq. 7), using UMAP (McInnes et al., 2018). Naive invariance regularization ($\mathcal{L}_{V0}$) leads to relatively non-discriminative representations, likely due to the model's pursuit of adversarial invariance; In fact, the cosine similarity between adversarial and clean features was 0.9985, highest among all methods in Tab. 5. In contrast, AR-AT learns more discriminative representations than the naive invariance regularization.

### 5.4 ON THE RELATION OF AR-AT TO KNOWLEDGE DISTILLATION (KD)-BASED DEFENSES

Here, we offer a novel perspective on the effectiveness of KD-based defenses, such as LBGAT and ARREST. In this section, the hyperparameters $\alpha$ and $\beta$ are set to 1.0, and the penultimate layer is used for regularization loss for simplicity.

**Using separate networks can relieve the "gradient conflict" and the mixture distribution problem.** Fig. 6 shows the impact of using "shared weights vs. separate networks" between adversarial and clean branches. We observe that *"Naive Inv. Loss (Eq. 3)"* and *"AR-AT w/o Split-BN (Eq. 6)"* struggle to attain high robustness when weights are shared but perform better with separate networks. We hypothesize that using separate networks (1) acts similarly to the stop-gradient operation by eliminating gradient flow between branches, and also (2) resemble split-BNs in avoiding the mixture distribution problem. In contrast, AR-AT (Eq. 7) maintains performance when layers are shared: Importantly, it even improves performance when layers are shared, indicating its

Table 6: **Comparison of AR-AT "with vs. without auto-balance (Eq. 8)"** of hyperparameter $\alpha$ and $\beta$ on CIFAR-10.

| $(\alpha, \beta)$ | ResNet-18 | | WRN-34-10 | |
|---|---|---|---|---|
| | Clean | PGD-20 | Clean | PGD-20 |
| $(1.0, 1.0)$ | 88.91 | 51.75 | 91.39 | 49.17 |
| $(1.0, 0.5)$ | 88.55 | 52.12 | 91.48 | 49.01 |
| $(1.0, 0.1)$ | 87.55 | 52.27 | 90.80 | 49.46 |
| $(0.5, 1.0)$ | 89.01 | 51.37 | 91.71 | 50.35 |
| $(0.1, 1.0)$ | 89.08 | 46.73 | 91.85 | 48.51 |
| **auto-bal.** | 87.93 | 52.13 | 90.81 | 52.72 |

Table 7: **Choice of regularized layers in AR-AT** on CIFAR-10.

| | Regularized layers | Clean | PGD-20 |
|---|---|---|---|
| ResNet | Penultimate layer (*"layer4"*) | 86.28 | 51.78 |
| | *"layer3", "layer4"* | 87.35 | 52.27 |
| | All BasicBlocks in *"layer4"* | 87.16 | 52.27 |
| | **All ReLUs in *"layer4"*** | **87.93** | **52.13** |
| WRN | Penultimate layer (*"layer3"*) | 88.87 | 52.50 |
| | *"layer2", "layer3"* | 88.95 | 50.74 |
| | All BasicBlocks in *"layer3"* | 89.50 | 51.76 |
| | **All ReLUs in *"layer3"*** | **90.81** | **52.72** |

superiority over KD-based regularization. This advantage of sharing layers may arise from more explicit invariance regularization compared to using a separate teacher network.

### 5.5 ABLATION STUDY

**Asymmetrizing logit-based regularization.** We also explore the benefits of an asymmetric structure for logit-based regularization, detailed in Appendix D. We find that TRADES also faces the "gradient conflict" and the mixture distribution problem, and *asymmetrizing* TRADES leads to significant improvements in robustness and accuracy, demonstrating the consistency of our findings.

**Effectiveness of hyperparameter auto-balance.** Tab. 6 shows an ablation study of AR-AT "with vs. without auto-balance (Sec. 3.4)" of the hyperparameters $\alpha$ and $\beta$, which controls the balance between adversarial and clean classification loss. The results demonstrate that auto-balancing performs comparable to the best hyperparameter manually obtained. Additionally, in the case of WRN-34-10, auto-balancing outperforms the best hyperparameter setting in Tab. 6, suggesting that it can discover superior hyperparameter configurations beyond manual tuning.

**Layer importance in invariance regularization.** Tab. 7 shows the ablation study of AR-AT on regularizing different levels of latent representations. ResNet-18 and WRN-34-10 have four and three *"layers"*, respectively. *"Layers"* are composed of multiple *"BasicBlocks"*, which consist of multiple convolutional layers followed by BN and ReLU. We show that regularizing multiple latent representations in the later stage of the network can achieve better performance. In practice, we recommend regularizing all ReLU outputs in the last stage of the network, which is the default of AR-AT. However, we note that the choice of layers for invariance regularization is not critical as long as the later-stage latent representations are regularized.

**Importance of batchNorm for robustness.** Tab. 8 presents the performance of AR-AT when the auxiliary BNs are used during inference. Intriguingly, despite sharing all the layers except BNs between $f_\theta$ and $f_\theta^{\text{auxBN}}$, using the auxiliary BNs during inference notably enhances the clean accuracy while compromising robustness significantly. This indicates that the auxiliary BN are exclusively tailored for clean inputs, highlighting the critical role of BNs in adversarial robustness.

Table 8: **Importance of BatchNorm.** We compare AR-AT on CIFAR10 using ResNet-18, using main BNs or auxiliary BNs during inference.

| Defense | Clean | PGD-20 |
|---|---|---|
| AR-AT ($f_\theta$) | 87.93 | 52.13 |
| AR-AT ($f_\theta^{\text{auxBN}}$) | 92.84 | 1.98 |

## 6 CONCLUSION

We explored invariance regularization-based adversarial defenses to mitigate robustness-accuracy trade-off. By addressing two challenges of the "gradient conflict" and the "mixture distribution problem," our method, AR-AT, achieves the state-of-the-art performance of mitigating the trade-off. The gradient conflict between classification and invariance loss is resolved using a stop-gradient operation, while the mixture distribution problem is mitigated through a split-BN structure. Additionally, we provide a new perspective on the success of KD-based methods, attributing it to resolution of these challenges. This paper provides a novel perspective to mitigate the robustness-accuracy trade-off.

**Limitations.** While AR-AT demonstrates the effectiveness of representation-level regularization, determining the appropriate layers for regularization remains ambiguous, especially for novel model architectures. Additionally, the proposed method may not be directly applicable to models without BNs, such as Transformers, which use Layer-Norm (Ba et al., 2016), and whether split-BN structure can be extended to Layer-Norm is an interesting future direction.

**Acknowledgments.** This work was partially supported by JSPS KAKENHI Grants JP21H04907 and JP24H00732, by JST CREST Grants JPMJCR18A6 and JPMJCR20D3 including AIP challenge program, by JST AIP Acceleration Grant JPMJCR24U3, and by JST K Program Grant JPMJKP24C2 Japan.

**Ethic statements.** We only use publicly available datasets, and do not involve any human subjects or personal data. Our work does not include harmful methodologies or societal consequences, but rather, aims to improve the robustness of machine learning models. We do not have any conflicts of interest or sponsorship to disclose. We have followed the ethical guidelines and research integrity standards in our work.

**Reproducability.** We provide the hyperparameters used in our experiments in Sec. 4. We also provide the additional implementation details of our method and the baseline methods in the appendix (Appendix A.2 and Appendix A.3). The code will be made available upon publication.

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

# A    ADDITIONAL EXPERIMENTS DETAILS

## A.1    DATASETS

Here, we provide the details of the datasets we used in our experiments. CIFAR10 and CIFAR100 (Krizhevsky & Hinton, 2009) are standard datasets for image classification with 10 and 100 classes, respectively, with a resolution of $32 \times 32$. Imagenette (Howard, 2019) is a subset of 10 easily classified classes from Imagenet (Deng et al., 2009). In our experiments, we used the version with the resolution of $160 \times 160$.

Table 9: The details of datasets we used in our experiments.

| Dataset | Resolution | Class Num. | Train | Val |
|---------|-----------|------------|-------|-----|
| CIFAR10 | $32 \times 32$ | 10 | 50,000 | 10,000 |
| CIFAR100 | $32 \times 32$ | 100 | 50,000 | 10,000 |
| Imagenette | $160 \times 160$ | 10 | 9,469 | 3,925 |

## A.2    EXPERIMENTS COMPUTE RESOURCES

In this work, we use NVIDIA A100 GPUs for our experiments. Training of AR-AT on CIFAR10 using ResNet-18 takes approximately 2 hours, and on CIFAR10 using WideResNet-34-10 takes approximately 10 hours.

## A.3    IMPLEMENTATION DETAILS OF BASELINE METHODS

Here, we describe the implementation details of the baseline defense methods.

- **Adversarial Training (AT)** (Madry et al., 2018): We use the simple implementation by Dongbin Na [3]. We aligned the hyperparameter setting described in the official GitHub repository [4].

- **TRADES** (Zhang et al., 2019): We use the official implementation [5] and used the default hyperparameter setting.

- **MART** (Wang et al., 2019): We use the official implementation [6] and used the default hyperparameter setting.

- **LBGAT** (Cui et al., 2021): We use the official implementation [7] and used the default hyperparameter setting. For Imagenette, we observed that the default learning rate causes gradient explosion, so we lowered the learning rate from 0.1 to 0.01. Note that LBGAT uses a ResNet-18 teacher network for both ResNet-18 and WRN-34-10 student networks.

- **ARREST** (Suzuki et al., 2023): They do not provide an official implementation, so we directly copied the results in Tab. 3 from their paper.

The comparison of loss functions is summarized in Tab. 10.

---

[3] https://github.com/ndb796/Pytorch-Adversarial-Training-CIFAR
[4] https://github.com/MadryLab/robustness
[5] https://github.com/yaodongyu/TRADES
[6] https://github.com/YisenWang/MART
[7] https://github.com/dvlab-research/LBGAT

Table 10: **Comparison of loss functions of defense methods.** $f_\theta(\cdot)$ is the predictions of the trained model, and $z_{(\theta)}$ and $z'_{(\theta)}$ is the latent representation of the model $f_\theta$ on clean and adversarial inputs, respectively. LBGAT and ARREST are knowledge distillation-based methods using a teacher model $g_\phi$. Our method AR-AT uses both clean and adversarial classification losses and employs representation-based regularization (Details in Sec. 3).

| Method | Classification Loss | | Regularization Loss |
|---|---|---|---|
| | Adversarial | Clean | |
| AT | $CE(f_\theta(x'), y)$ | | |
| TRADES | | $CE(f_\theta(x), y)$ | $KL(f_\theta(x)\|f_\theta(x'))$ |
| MART | $BCE(f_\theta(x'), y)$ | | $KL(f_\theta(x)\|f_\theta(x')) \cdot (1 - f_\theta(x)_y)$ |
| LBGAT | | $CE(g_\phi(x), y)$ | $MSE(f_\theta(x'), g_\phi(x))$ |
| ARREST | $CE(f_\theta(x'), y)$ | | $CosSim(z'_{(\theta)}, z_{(\phi)})$, where $g_\phi$ is a fixed std. model |
| AR-AT (ours) | $CE(f_\theta(x'), y)$ | $CE(f_\theta^{\text{auxBN}}(x), y)$ | $CosSim(h(z'_{(\theta)}), \text{sg}(z_{(\theta^{\text{auxBN}})}))$ |

# B  ADDITIONAL PERFORMANCE COMPARISONS

## B.1  COMPARISON WITH STATE-OF-THE-ART DEFENSE METHODS

Table 11: Comparison of clean accuracy and robust accuracy against AutoAttack (AA) for **WideResNet-34-10 trained on CIFAR10**. The compared methods are sorted by the sum of clean and robust accuracy.

| Method | Clean | AutoAttack | Sum. | Reference |
|---|---|---|---|---|
| AT (Madry et al., 2018) | 86.06 | 46.26 | 132.32 | Reproduced |
| FAT (Zhang et al., 2020) | 89.34 | 43.05 | 132.39 | Copied from (Suzuki et al., 2023) |
| MART (Wang et al., 2019) | 86.10 | 49.11 | 135.21 | Reproduced |
| TRADES (Zhang et al., 2019) | 84.33 | 51.75 | 136.08 | Reproduced |
| NuAT2 (Sriramanan et al., 2021) | 84.76 | 51.27 | 136.03 | Copied from the original paper |
| IAD (Zhu et al., 2021) | 85.09 | 52.29 | 137.38 | Copied from the original paper |
| TRADES+Rand Jin et al. (2023) | 85.51 | 54.00 | 139.51 | Copied from the original paper |
| AWP (Wu et al., 2020) | 85.57 | 54.04 | 139.61 | Copied from the original paper |
| S$^2$O (Jin et al., 2022) | 85.67 | 54.10 | 139.77 | Copied from the original paper |
| ARREST (Suzuki et al., 2023) | 90.24 | 50.20 | 140.44 | Copied from the original paper |
| LBGAT (Cui et al., 2021) | 88.28 | 52.49 | 140.77 | Reproduced |
| NuAT2+WA (Sriramanan et al., 2021) | 86.32 | 54.76 | 141.08 | Copied from the original paper |
| AT+HF (Li et al., 2024) | 87.53 | 55.58 | 143.11 | Copied from the original paper |
| TRADES+AWP+Rand (Jin et al., 2023) | 86.10 | **57.10** | 143.20 | Copied from the original paper |
| **AR-AT (ours)** | **90.81** | 50.77 | 141.66 | (Ours) |
| **AR-AT+SWA (ours)** | 90.06 | 54.03 | **144.09** | (Ours) |

Table 12: Comparison of clean accuracy and robust accuracy against AutoAttack (AA) for **WideResNet-34-10 trained on CIFAR100**. The compared methods are sorted by the sum of clean and robust accuracy.

| Method | Clean | AutoAttack | Sum. | Reference |
|---|---|---|---|---|
| MART (Wang et al., 2019) | 57.75 | 24.89 | 82.64 | Reproduced |
| AT (Madry et al., 2018) | 59.83 | 23.94 | 83.77 | Reproduced |
| TRADES (Zhang et al., 2019) | 57.61 | 26.88 | 84.49 | Reproduced |
| FAT (Zhang et al., 2020) | 65.51 | 21.17 | 86.68 | Copied from (Suzuki et al., 2023) |
| AT+HF (Li et al., 2024) | 58.99 | 28.65 | 87.64 | Copied from the original paper |
| IAD (Zhu et al., 2021) | 60.72 | 27.89 | 88.61 | Copied from the original paper |
| TRADES+HF (Li et al., 2024) | 58.70 | **30.29** | 88.99 | Copied from the original paper |
| AWP (Wu et al., 2020) | 60.38 | 28.86 | 89.24 | Copied from the original paper |
| TRADES+Rand Jin et al. (2023) | 62.93 | 27.90 | 90.83 | Copied from the original paper |
| $S^2O$ (Jin et al., 2022) | 63.40 | 27.60 | 91.00 | Copied from the original paper |
| TRADES+AWP+Rand (Jin et al., 2023) | 64.71 | 30.20 | 94.91 | Copied from the original paper |
| LBGAT (Cui et al., 2021) | 69.26 | 27.53 | 96.79 | Reproduced |
| ARREST (Suzuki et al., 2023) | **73.05** | 24.32 | 97.37 | Copied from the original paper |
| **AR-AT (ours)** | 72.23 | 24.97 | 97.20 | (Ours) |
| **AR-AT+SWA (ours)** | 72.41 | 27.15 | **99.56** | (Ours) |

We provide more comparison with state-of-the-art defense methods in Tab. 11 and Tab. 12 We observe that AR-AT achieves high clean accuracy while maintaining high robustness. These results demonstrate the effectiveness of AR-AT in mitigating the robustness-accuracy trade-off.

## B.2    TRADE-OFF PLOT WITH ARDIST

In this section, to visualize the trade-off between robustness and accuracy, we plot them for AR-AT and the baseline methods. We also added ARDist Suzuki et al. (2023), which uses existing methods to approximate a curve representing the accuracy-robustness trade-off. Methods positioned towards the top-right are better, as they achieve both high robustness and accuracy. Our method, ARAT, consistently appears on the top-right side across most scenarios. The absence of the ARDist curve in the ResNet-18 plots is due to it being positioned far in the top-right corner, exceeding the plot's range. This occurs because the ARDist curve is calculated using the best models for each method, typically from the WideResNet family.

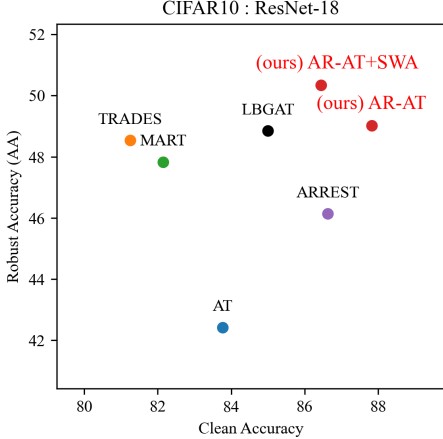

Figure 7: Trade-off plot on CIFAR10 with ResNet-18.

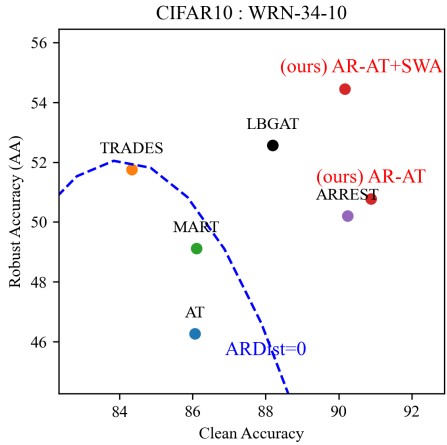

Figure 8: Trade-off plot on CIFAR10 with WideResNet-34-10.

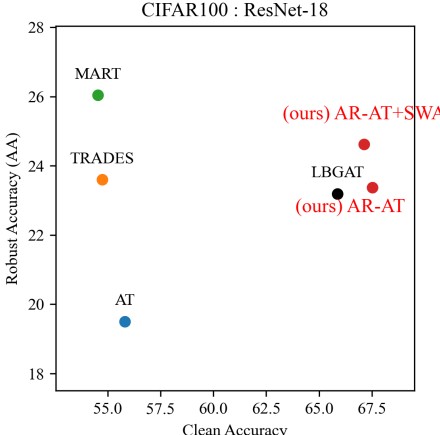

Figure 9: Trade-off plot on CIFAR100 with ResNet-18.

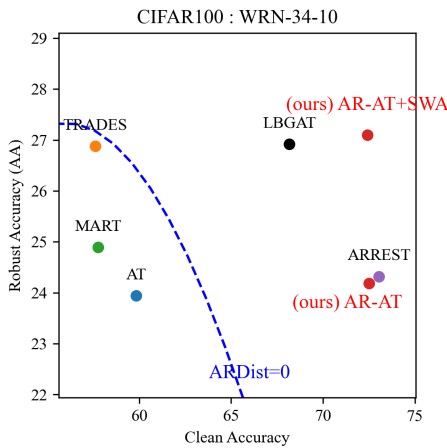

Figure 10: Trade-off plot on CIFAR100 with WideResNet-34-10.

## B.3 ADJUSTING TRADE-OFF IN AR-AT

To adjust the robusness-accuracy trade-off in AR-AT, the streightforward way is to change the perturbation size of AEs ($\epsilon$) in the adversarial training. By adjusting the perturbation size, in Fig. 13, we show that the trade-off between clean accuracy and robust accuracy can be adjusted. We show that by adjusting the perturbation size, AR-AT achieves both higher clean accuracy and robust accuracy than LBGAT, and TRADES-AWP.

Table 13: Trade-off between clean accuracy and robust accuracy by adjusting the perturbation size for adversarial training ($\epsilon$) in AR-AT. In the CIFAR10/WideResNet-34-10 setting, we outperform LBGAT and TRADES-AWP for both clean and robust accuracy.

| Dataset | Model | Method | eps (x/255) | Clean | AA |
|---|---|---|---|---|---|
| CIFAR10 | WRN-34-10 | LBGAT | 8 (default) | 88.28 | 52.49 |
| | | TRADES-AWP | 4 | 89.36 | 50.67 |
| | | | 5 | 87.76 | 52.63 |
| | | | 6 | 86.79 | 53.92 |
| | | | 8 (default) | 84.88 | 55.35 |
| | | ARAT | 8 (default) | 90.81 | 51.19 |
| | | | 9 | 89.83 | 52.60 |

## B.4 $L_2$-BOUNDED ADVERSARIAL TRAINING

We provide evidences that AR-AT also works in $L_2$-bounded adversarial training scenarios. The perturbation size is set to $\epsilon = 0.5$ for $L_2$-bounded adversarial training. Compared with the baseline $L_2$-bounded adversarial training (AT*), our proposed method, ARAT, shows better clean accuracy and robust accuracy against AA on both CIFAR10 and CIFAR100 datasets.

Table 14: Comparison of $L_2$-bounded adversarial training on CIFAR10 and CIFAR100 datasets. We compare the clean accuracy and robust accuracy against AutoAttack (AA).

| Dataset | Model | Method | Clean | AA |
|---|---|---|---|---|
| CIFAR10 | PreAct ResNet-18 | AT* | 88.91 | 65.93 |
| | ResNet-18 | AR-AT | 92.10 | 68.25 |
| CIFAR100 | PreAct ResNet-18 | AT* | 60.50 | 35.27 |
| | ResNet-18 | AR-AT | 73.86 | 37.92 |

## B.5 ABLATION STUDY: CONTRIBUTIONS OF EACH COMPONENT IN AR-AT

Tab. 15 shows the abltation study of three components in AR-AT: stop-gradient operation, predictor MLP, and split-BN. The results demonstrate that Split-BN consistently improves the robustness and accuracy compared to the naive invariance regularization (i.e., (1) in Tab. 15) by mitigating the mixture distribution problem. On the other hand, we observe that mitigation of "gradient conflict" does not necessarily improve the robustness and accuracy (i.e., (2) and (4) in Tab. 15): it is explicitly effective when the mixture distribution problem is resolved by split-BN (i.e., (8) in Tab. 15). Therefore, we conclude that resolving both issues of "gradient conflict" and mixture distribution problem is important to achieve high robustness and accuracy.

Table 15: **Contributions of each component in AR-AT**, for ResNet-18 trained on CIFAR10. "Grad-sim." represents the average gradient similarity between the classification loss and invariance loss with respect to $\theta$ during training. Here, the penultimate representation is regularized.

| | Method | | | | | |
| | Stop-grad | Pred. | Split-BN | Clean | PGD-20 | Grad-sim. |
|---|---|---|---|---|---|---|
| (1) $\mathcal{L}_{V0}$ | | | | 82.93 | 48.89 | 0.06 |
| (2) $\mathcal{L}_{V1}$ | ✓ | | | 82.47 | 48.57 | 0.68 |
| (3) | | ✓ | | 83.49 | 48.22 | 0.00 |
| (4) $\mathcal{L}_{V2}$ | ✓ | ✓ | | 83.00 | 49.14 | 0.63 |
| (5) | | | ✓ | 84.35 | 51.34 | 0.14 |
| (6) | ✓ | | ✓ | 85.51 | 51.46 | 0.59 |
| (7) | | ✓ | ✓ | 84.07 | 50.79 | 0.00 |
| (8) $\mathcal{L}_{V3}$ | ✓ | ✓ | ✓ | **86.29** | **52.40** | 0.52 |

### B.6 HYPERPARAMETER STUDY: STRENGTH OF INVARIANCE REGULARIZATION $\gamma$

Tab. 16 shows the ablation study of AR-AT on the strength of invariance regularization $\gamma$. We observe that the performance is not too sensitive to the strength of invariance regularization $\gamma$. Nevertheless, the optimal $\gamma$ depends on the architecture. For example, $\gamma = 30.0$ is optimal for ResNet-18, while $\gamma = 100.0$ is optimal for WRN-34-10.

Table 16: AR-AT's sensitivity to the strength of invariance regularization $\gamma$.

| | $\gamma$ | CIFAR10 Clean | CIFAR10 PGD-20 | CIFAR100 Clean | CIFAR100 PGD-20 |
|---|---|---|---|---|---|
| ResNet-18 | 10.0 | 87.90 | 51.53 | 66.89 | **26.79** |
| | **30.0 (default)** | **87.93** | **52.13** | **68.07** | 26.76 |
| | 50.0 | 87.54 | 51.27 | 67.72 | 26.52 |
| | 100.0 | 87.06 | 50.14 | 66.19 | 26.02 |
| | 120.0 | 86.46 | 50.15 | 65.31 | 25.71 |
| WRN-34-10 | 10.0 | 89.21 | 48.71 | 67.46 | 26.86 |
| | 30.0 | 90.93 | 49.68 | 70.69 | 26.86 |
| | 50.0 | 91.36 | 50.51 | 71.52 | 26.46 |
| | **100.0 (default)** | 91.29 | **52.24** | 71.58 | **28.06** |
| | 120.0 | **91.39** | 51.94 | **71.92** | 27.31 |

## C ADDITIONAL EXPLANATION ON GRADIENT CONFLICT

In addition to the main text, we provide a step-by-step explanation on why the second term of Eq. 5, $Dist(sg(z'), z)$, causes gradient conflict.

**1. Adversarial Representations are Perturbed:** Adversarial representations $z'$ are perturbed from clean representation $z$ due to input perturbations $\delta$. For example, with a single linear layer of weights $W$,

$$z = W^T x, \tag{9}$$
$$z' = W^T(x + \delta) = W^T x + W^T \delta = z + W^T \delta. \tag{10}$$

The gap between $z$ and $z'$ tends to increase with deeper layers.

**2. Role of Stop-Gradient (sg):** The stop-gradient operation treats representations as constant in the invariance regularization loss.

**3. Minimizing the first term, $Dist(z', sg(z))$:** This aligns adversarial representations $z'$ closer to clean representations $z$. Since $z'$ is corrupted with $W^T \delta$, minimizing this term "purifies" the representation by reducing $W^T \delta$ to zero, mitigating the impact of adversarial noise $\delta$ on the learned representations. This ensures that meaningless input-space perturbations do not alter the underlying semantics.

**4. Minimizing the second term, $Dist(sg(z'), z)$:** This aligns clean representations $z$ closer to corrupted adversarial representations $z'$. Specifically, $z$ is pulled toward $z' = z + W^T \delta$, where $W^T \delta$ is unpredictable from $z$ due to dependence on unknown $\delta$. This "contaminates" $z$, resulting in the unnecessary corruption of learned representations.

**5. Conclusion:** Minimizing $Dist(sg(z'), z)$ can harm classification objectives by corrupting learned representations. This leads to a conflict between clean and adversarial objectives.

# D ASYMMETRIZING LOGIT-BASED REGULARIZATION: IMPROVING TRADES AND ITS RELATION TO LBGAT

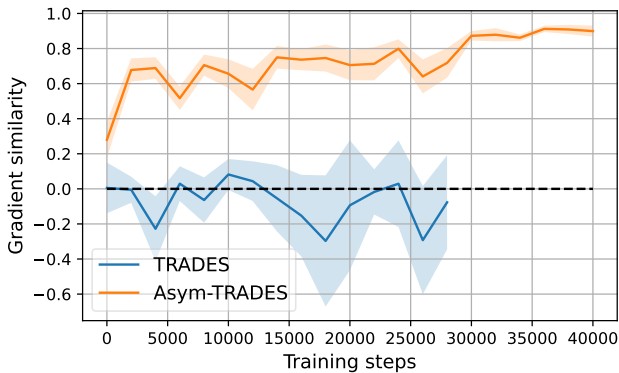

Figure 11: **Comparison of gradient similarity between "TRADES vs. Asymmetrized-TRADES (Asym-TRADES)"**. We compare gradient similarity between the classification loss and invariance loss with respect to $\theta$ during training.

The main text mainly focused on employing invariance regularization on latent representations to mitigate the robustness-accuracy trade-off. Thus, this section discusses logit-based invariance regularization, such as TRADES (Zhang et al., 2019) and LBGAT (Cui et al., 2021), and shows that similar discussion to representation-based regularization methods is applicable for logit-based regularization.

**TRADES also suffers from "gradient conflict."** In Fig. 11, we visualize the gradient similarity between the classification loss and invariance loss with respect to $\theta$ during training. TRADES also suffers from "gradient conflict," where the gradient similarity between the classification loss is negative for many layers. This can be explained by the difficulty in achieving prediction invariance while maintaining discriminative ability, which is the same reason why the naive invariance regularization (Eq. 3) suffers from "gradient conflict" (Sec. 3.1).

Table 17: Comparison of loss functions between TRADES, Asymmetrized-TRADES, and LBGAT. LBGAT is similar to TRADES, except that it employs a separate teacher network to regularize the student network.

| Method | Classification Loss | | Regularization Loss |
|---|---|---|---|
| | Adversarial | Clean | |
| TRADES | | $CE(f_\theta(x), y)$ | $KL(f_\theta(x) \| f_\theta(x'))$ |
| Asym-TRADES | $CE(f_\theta(x'), y)$ | $CE(f_\theta^{\text{auxBN}}(x), y)$ | $KL(f_\theta(x') \| \text{sg}(f_\theta^{\text{auxBN}}(x)))$ |
| LBGAT | | $CE(g_\phi(x), y)$ | $MSE(f_\theta(x'), g_\phi(x))$ |

Table 18: **Comparison of TRADES vs. Asym-TRADES** for ResNet18 trained on CIFAR10.

| Method | Clean | AutoAttack |
|---|---|---|
| TRADES | 81.25 | 48.54 |
| Asym-TRADES | 85.62 | 48.58 |
| w/o Stop-grad | 82.13 | 48.97 |
| w/o Split-BN | 80.90 | 49.93 |
| LBGAT | 85.50 | 49.26 |

**Aymmetrizing TRADES with stop-gradient operation and split-BN.** To further validate our findings, we consider asymmetrizing TRADES based on AR-AT's strategy. Specifically, we replace the regularization term of AR-AT with TRADES's regularization term, which we call "Asym-TRADES," as shown in Tab. 17. Here, we did not employ the predictor MLP in Asym-TRADES since having a bottleneck-structured predictor MLP does not make sense for logits. Similar to the phenomenon in representation-based regularization, Asym-TRADES does not suffer from "gradient conflict" (Fig. 11), and achieves higher robustness and accuracy than TRADES (Tab. 18).

**Asymmetrized TRADES achieve approximately the same performance as LBGAT.** Intriguingly, Tab. 18 demonstrates that Asym-TRADES achieve approximately the same performance as LBGAT. This validates our hypothesis that the effectiveness of the KD-based method LBGAT can be attributed to resolving "gradient conflict" and the mixture distribution problem, as discussed in Sec. 5.4. LBGAT employs a separate teacher network to regularize the student network, which implicitly resolves the "gradient conflict" and the mixture distribution problem.

**Representation-based regularization can outperform logit-based regularization** By comparing Tables 18 and 3, we observe that AR-AT with representation-based regularization can outperform AR-AT with logit-based regularization, "Asym-TRADES". We hypothesize that, since early layers have more diverse levels of information than logits, representation-based regularization can be more effective than logit-based regularization.

# E  GRADIENT CONFLICT BEYOND MINI-BATCH LEVEL: ANALYSIS ACROSS BATCH SIZES.

To investigate whether the gradient conflict observed at the mini-batch level reflects conflicts in the full input distribution, we provide empirical results showing that gradient conflict persists across different batch sizes.

In Figure 12, we plot the proportion of parameters experiencing gradient conflict across varied batch sizes using ResNet-18 trained on CIFAR-10. With the naive invariance loss ($L_{V0}$), the conflict ratio remains consistently high, with around 50% of parameters experiencing gradient conflict, regardless of the batch size. Actually, the conflict ratio appears to increase with the larger batch size. This indicates that the gradient conflict is not limited to mini-batch levels but is prevalent throughout the entire input distribution.

Thus, we conclude that the gradient conflict can be the fundamental issue affecting the robustness-accuracy trade-off when using invariance regularization.

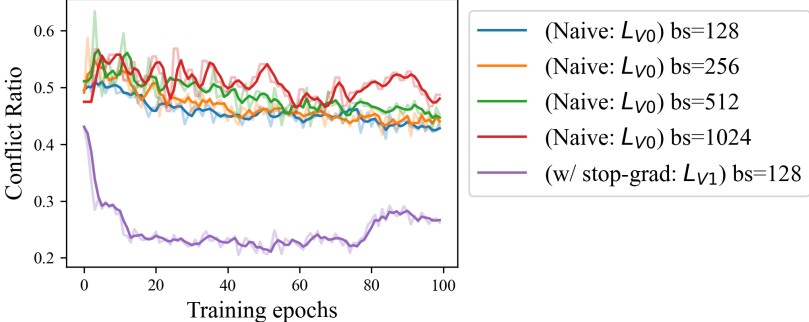

Figure 12: **Proportion of parameters experiencing gradient conflict across varied batch sizes ("bs").** With the naive invariance loss ($L_{V0}$), the conflict ratio remains consistently high across different batch sizes, indicating that the gradient conflict is not limited to mini-batch levels but is prevalent throughout the entire input distribution.

## F ADDITIONAL ANALYSIS ON MIXTURE DISTRIBUTION PROBLEM

### F.1 FURTHER ANALYSIS OF $L2\ norm(z' - z)$

In Figure 13, we show the L2 distance between adversarial and clean features ($||z' - z||_2$) for ResNet-18 trained on CIFAR-10. In addition to Figure 3 from the main text, we include results for standard training and adversarial training (AT) without invariance regularization for comparison.

**All methods show increasing in $||z' - z||_2$ over time, reflecting the robustness-accuracy trade-off.** As the model accuracy improves, adversarial perturbations become easier to find, leading to higher $||z' - z||_2$. This poses a challenge for methods using both clean and adversarial inputs during training: the diverging distributions of clean and adversarial samples make it difficult for BatchNorm (BN) layers to correctly estimate batch statistics, leading to the mixture distribution problem. Note that standard training and AT avoid this issue by using only clean or adversarial samples, respectively.

**Stop-grad increases $||z' - z||_2$ compared to AR-AT without it.** While stop-grad resolves gradient conflict, it weakens the invariance regularization by removing the second term in Eq. 5 ($(Dist(z', sg(z)) + Dist(z, sg(z')))/2$). This leads to an increase in $||z' - z||_2$, exacerbating the mixture distribution problem. Our split-BN strategy effectively mitigates this issue, enhancing the utility of stop-grad.

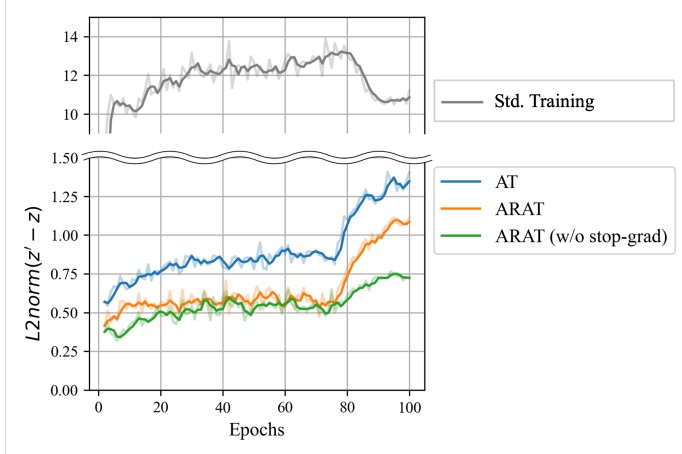

Figure 13: L2 distance between adversarial and clean features ($||z' - z||_2$). All training methods show an increase in $||z' - z||_2$ over time, reflecting the robustness-accuracy trade-off. Using stop-grad increases $||z' - z||_2$ compared to AR-AT without stop-grad, exacerbating the mixture distribution problem.

### F.2 SPLIT-BN STABILIZES THE BATCHNORM STATISTICS

To further evaluate the effectiveness of Split-BN, we analyze the stability of the BatchNorm (BN) statistics during training.

BN normalizes each input $x_i$ using the running estimates of the mean and variance from mini-batches, as follows:

$$\hat{x}_i \leftarrow \frac{x_i - \mu}{\sqrt{\sigma^2 + \epsilon}}, \tag{11}$$

where the moving averages of the mini-batch mean $\mu$ and variance $\sigma$ are updated with a momentum term:

$$\mu \leftarrow \text{momentum} * \mu + (1 - \text{momentum}) * \mu_\mathcal{B}, \quad \mu_\mathcal{B} = \frac{1}{m} \sum_{i=1}^{m} x_i, \tag{12}$$

$$\sigma^2 \leftarrow \text{momentum} * \sigma^2 + (1 - \text{momentum}) * \sigma_\mathcal{B}^2, \quad \sigma_\mathcal{B}^2 = \frac{1}{m} \sum_{i=1}^{m} (x_i - \mu_\mathcal{B})^2. \tag{13}$$

In Figure 14, we plot the variance of $\mu$ of the BN layers computed at each epoch. With the naive invariance loss ($L_{V0}$), the variance of the $\mu$ remains high throughout training. In contrast, split-BN stabilizes the updates of BN statistics by separating clean and adversarial BN statistics. Specifically, (1) the variance of adversarial BN statistics is comparable or lower than that of the shared BN without Split-BN, and (2) the variance for clean BN statistics is consistently lower. This aligns with our intuition, as clean samples are unaffected by the dynamic nature of adversarial perturbations during training. These results confirm that the split-BN stabilizes the estimation of BN statistics, effectively addressing the mixture distribution problem.

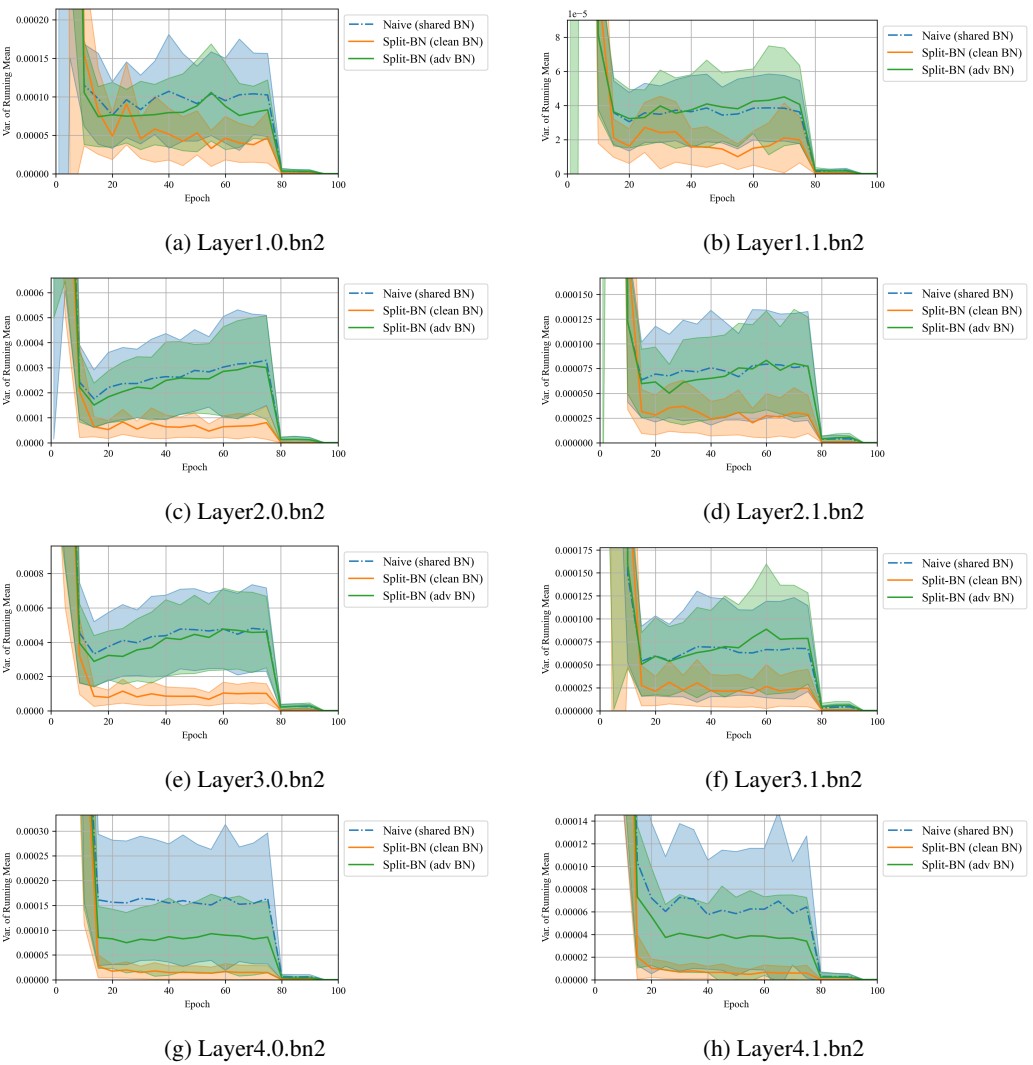

Figure 14: **Variance of running means of the BatchNorm (BN) layers of ResNet-18.** We compute the variance at each epoch and plot the results. With the naive invariance loss ($L_{V0}$), the variance of the running means remains high throughout training. In contrast, applying Split-BN stabilizes both the clean and adversarial BN statistics, leading to lower variance.

## G  COMPUTATIONAL TIME

In Table 19, we report the computational time of the baseline methods and ARAT, trained on CIFAR-10 with ResNet-18 using a single NVIDIA A100 GPU and a batch size 128.

**AR-AT is faster than TRADES and LBGAT.** AR-AT is faster than TRADES and LBGAT, because these methods use the $KL$ divergence ($KL(f_\theta(x)||f_\theta(x')))$ for adversarial example generation,

requiring forward passes for both clean and adversarial images, while AR-AT uses the cross-entropy loss ($CE(f_\theta(x'), y)$), which only requires forwarding the adversarial images.

Compared to AT, AR-AT introduces a 15% increase in time. This is due to (1) the need for forward passes of both clean and adversarial images for loss calculation (please refer Table 10), and (2) additional updates for separate BatchNorm layers and the predictor MLP head.

Table 19: Computational time of AR-AT and baseline methods, trained on CIFAR-10 using ResNet-18, along with clean and robust (AutoAttack; AA) accuracies.

| Method | Clean | AA | Time (sec/batch) | Time (rel. to AT) |
|---|---|---|---|---|
| AT | 83.77 | 42.42 | $0.122 \pm 0.045$ | 1 |
| TRADES | 81.25 | 48.54 | $0.184 \pm 0.112$ | 1.51 |
| LBGAT | 85.00 | 48.85 | $0.183 \pm 0.200$ | 1.50 |
| (ours) AR-AT | 87.82 | 49.02 | $\underline{0.140 \pm 0.039}$ | $\underline{1.15}$ |

