# OpenReview forum: "Rethinking Invariance Regularization in Adversarial Training to Improve Robustness-Accuracy Trade-off"
_ICLR.cc/2025/Conference — ICLR 2025 Poster_

### Official Review · Reviewer_zuTF · 2024-10-27

**Soundness:** 3
**Presentation:** 3
**Contribution:** 3
**Rating:** 6
**Confidence:** 4

**Summary:**

This work addresses the fundamental robustness-accuracy trade-off in adversarial defenses through a novel invariance regularization approach. They proposed the method, AR-AT, achieves state-of-the-art performance by systematically addressing two key challenges: the gradient conflict between classification and invariance losses, and the mixture distribution problem in adversarial training. They resolved the gradient conflict through a strategic stop-gradient operation, while implementing a split batch normalization structure to handle the mixture distribution challenge.

**Strengths:**

This work is well-written and easy to understand. The perspective on gradient conflict is particularly interesting, connecting previously disparate threads in adversarial robustness research.

**Weaknesses:**

First, the paper's central premise about gradient conflicts requires deeper theoretical examination. Similar gradient conflicts arise in various scenarios, such as when optimizing the same loss across different mini-batches, yet these conflicts don't necessarily impact model generalization (or adversarial robustness). The paper doesn't provide sufficient theoretical justification for why resolving gradient conflicts specifically improves adversarial robustness.

Second, there's a critical gap in the analysis between local optimization dynamics and global distributional properties. The gradient conflicts observed at the mini-batch level may not accurately reflect the underlying conflicts in the full input distribution. Without this theoretical bridge, it's unclear whether the proposed solution addresses the fundamental cause of the robustness-accuracy trade-off.

Third (minor), the experimental results may not be the SOTA. Ref https://github.com/wzekai99/DM-Improves-AT

**Questions:**

Ref Weaknesses.

---

> ### Author Response · Authors · 2024-11-19
> **Rebuttal by Authors (1/2)**
>
> We sincerely appreciate your recognition that (1) this work is well-written and easy to understand, and (2) the perspective on gradient conflict is particularly interesting, as it connects previously disparate threads in adversarial robustness research.
>
> Below, we address the concerns one by one.
>
> ---
> ## [W1: Deeper theoretical examination on gradient conflicts]
> We appreciate the reviewer’s comment and will carefully address the points raised.
>
> **[Clarification on the definition of "gradient conflict"]**
>
> As noted, there are two distinct types of "conflicts" in optimization:
> - (1) **Gradient conflicts between multiple objectives (i.e., $\nabla_{\theta} L1 \cdot \nabla_{\theta} L2$ < 0)**, which arise when simultaneously optimizing multiple objectives. *This is the focus of our work.* We have clarified this in Sec. 3.1.
> - (2) **Conflicts across mini-batches (i.e., $\nabla_{\theta} L_t \cdot \nabla_{\theta} L_{t+1} < 0$)**, which result from inconsistent gradients between consecutive mini-batches. *This is beyond the scope of our work.*
>
> While both can cause oscillations and hinder convergence, they require different solutions.
> For (2), design choices like batch size, momentum in the optimizer, and learning rate scheduling can help mitigate batch-wise conflicts.
> In contrast, addressing (1) involves specific adjustments in the multi-objective optimization, such as altering loss functions, which is the focus of our approach.
>
> **[Clarification on our claims and justifications regarding gradient conflict]**
>
> > The paper doesn't provide sufficient theoretical justification for why resolving gradient conflicts specifically improves adversarial robustness.
>
> Sorry for any confusion.
> To clarify, **we did not intend to suggest that resolving gradient conflict directly improves robustness.**
> Our argument is that **resolving gradient conflict improves the robustness-accuracy trade-off** in adversarial training with invariance regularization, **mainly by enhancing model accuracy without compromising robustness**. This is demonstrated in Table 1.
>
>
> Below, we summarize how we verified our claims for easier understanding:
> - **Known fact:** *Resolving gradient conflict generally leads to better multi-loss optimization.* This is supported by prior works in multi-task learning and domain generalization (please refer to Section 3.1).
> - **Claim 1:** *Gradient conflict exists in adversarial training with invariance regularization.*
>    - **Justification:** Empirical evidence provided in Figure 2.
> - **Claim 2:** *In the naive invariance loss, $Dist(z', z) = (Dist(z', sg(z)) + Dist(sg(z'), z)) / 2$,  the second term $Dist(sg(z'), z)$ causes gradient conflict. Using stop-grad (only minimizing $Dist(z', sg(z))$) resolves this conflict.*
>    - **Justification:** Empirical evidence in Figure 2 shows that the term $Dist(sg(z'), z)$ induces gradient conflict. *We have updated Figure 2 based on the reviewer JoTP's suggestion.*
> - **Claim 3:** *When gradient conflict is resolved, the robustness-accuracy trade-off is mitigated.*
>    - **Justification:** Supported by empirical results in Table 1.
>
> We believe that our claims are effectively verified, and our empirical validation is both clear and convincing, as acknowledged by other reviewers (e.g., “in-depth analysis” by daZ8, “Experiments clearly show the effectiveness of each technique” by Kdc6). Nonetheless, we acknowledge that a deeper theoretical analysis would be valuable and consider this a promising direction for future work.
> We highly appreciate your valuable feedback.
>
> ---
> ## [W2: Analysis between local optimization dynamics and global distributional properties]
>
> We highly appreciate your constructive suggestion.
> To investigate whether the gradient conflict observed at the mini-batch level reflects conflicts in the full input distribution, we provide empirical results showing that gradient conflict persists across different batch sizes.
>
> In Figure 12 (Appendix H), we plot the proportion of parameters experiencing gradient conflict across varied batch sizes. With the naive invariance loss ($L_{V0}$), the conflict ratio remains consistently high, with around 50\% of parameters experiencing gradient conflict, regardless of the batch size.
> Actually, the conflict ratio appears to increase with the larger batch size.
> This indicates that the gradient conflict is not limited to the mini-batch level but is prevalent throughout the entire input distribution.

---

> > ### Author Response · Authors · 2024-11-19
> > **Rebuttal by Authors (2/2)**
> >
> > ## [W3: (minor) the experimental results may not be the SOTA]
> >
> > The paper (https://github.com/wzekai99/DM-Improves-AT) proposes adversarial training with a large number of synthetic images generated by a Diffusion model.
> > In our paper, we focused on the original datasets with real-world images.
> >
> > It is indeed an important future direction to extend the use of Diffusion-generated images for all baseline methods. However, this would require significant computational resources and additional hyperparameter tuning, making it infeasible to include in the rebuttal phase.
> >
> > Nevertheless, Appendix B provides more comparisons with various methods using only real-world data.
> > Tables 11 and 12 show that AR-AT achieves state-of-the-art performance on CIFAR-10 and ranks second on CIFAR-100, further reinforcing the results presented in the main text.
> >
> > ---
> > Feel free to ask any further questions if anything is unclear. If you feel your concerns have been adequately addressed, we would appreciate it if you could adjust the score accordingly.

---

> ### Comment · Reviewer_zuTF · 2024-11-24
> **Response by reviewer**
>
> Thank you for your thorough response, and sorry for my late response. The revised manuscript provides better clarity on the key concepts. While the work shows considerable merit and I would be open to increasing my score, there are two fundamental questions that warrant deeper discussion:
>
> 1. Regarding Claim 2, could you elaborate on the theoretical (or intuitive) foundation of gradient conflict arising from the second term? A more rigorous theoretical analysis or intuitive explanation would strengthen this crucial aspect of your argument beyond the empirical observations.
> 2. The connection between addressing gradient conflict and mitigating the trade-off requires further elaboration. Specifically, the **trade-off on the test set** involves complex interactions between generalization and robust generalization. Could you provide a more thorough (not empirical) discussion explaining this relationship?
>
> While your experimental results are promising, I believe additional theoretical insights into these "why" questions would significantly strengthen the paper's contribution and its suitability for ICLR. Such additions would help bridge the gap between empirical findings and theoretical/intuitive understanding.

---

> > ### Author Response · Authors · 2024-11-25
> > **Response by Authors**
> >
> > Thank you for your response and for taking the time to review our manuscript.
> >
> > We are pleased to hear that the paper's clarity has improved and sincerely appreciate your recognition of its strengths and consideration of raising the score.
> >
> > We are also deeply grateful for your insightful questions and the effort you invested in raising them. We totally agree that these questions are highly important for a deeper understanding of the observed phenomena, and we will address them as carefully and comprehensively as possible.
> >
> > ## [1: Theoretical (or intuitive) foundation of gradient conflict arising from the second term $Dist(sg(z'), z)$]
> >
> > Here, we provide an intuitive explanation, step-by-step:
> >
> > **1. Adversarial Representations are Perturbed:**
> >
> > - Adversarial representations $z'$ are perturbed from clean representation $z$ due to input perturbations $\delta$. For example, with a single linear layer of weights $W$,
> > \begin{aligned}
> >   z = W^T x,
> > \end{aligned}
> > \begin{aligned}
> >   z' = W^T(x + \delta) = W^T x + W^T \delta = z + W^T \delta.
> > \end{aligned}
> > - The gap between $z$ and $z'$ tends to increase with deeper layers.
> >
> > **2. Role of Stop-Gradient (sg):**
> > - The stop-gradient operation treats representations as constant in the invariance regularization loss.
> >
> > **3.  Minimizing the first term, $Dist(z', sg(z))$:**
> > - This aligns adversarial representations $z'$ closer to clean representations $z$. Since $z'$ is corrupted with $W^T \delta$, **minimizing this term "purifies" the representation by reducing $W^T \delta$ to zero**, mitigating the impact of adversarial noise $\delta$ on the learned representations. This ensures that meaningless input-space perturbations do not alter the underlying semantics.
> >
> > **4. Minimizing the second term, $Dist(sg(z'), z))$:**
> > - This aligns clean representations $z$ closer to corrupted adversarial representations $z'$. **Specifically, $z$ is pulled toward $z' = z + W^T \delta$, where $W^T \delta$ is unpredictable from $z$ due to dependence on unknown $\delta$. This "contaminates" $z$**, resulting in the unnecessary corruption of learned representations.
> >
> > **5. Conclusion:**
> > - Minimizing $Dist(sg(z'), z)$ can harm classification objectives by corrupting learned representations. This leads to a **conflict** between clean and adversarial objectives.
> >
> > ---
> > ## [2. Relationship between gradient conflict and trade-off on the test set (generalization and robust generalization)]
> >
> > In short, resolving gradient conflict improves both clean and robust accuracy by enhancing optimization stability and convergence.
> > It helps harmonize the tasks of classification and invariance, boosting both clean and robust generalization.
> >
> > Here, we explain in detail:
> > - **(Fact 1) Resolving gradient conflict improves multi-loss representation learning.**
> > 	- It ensures stable training and effective feature learning in multi-loss optimization. *Without resolution, one objective may dominate, causing suboptimal feature learning.*
> > - **(Fact 2) Adversarial training (AT) with invariance regularization improves both clean and robust generalization**, as shown in existing studies.
> > - **(Our contribution) We identified gradient conflict in AT with invariance regularization. Resolving it enhances both clean and robust generalization.**
> >    - **Generalization (Clean Accuracy):** Resolving gradient conflict prevents invariance regularization from overpowering classification gradients, preserving clean representations and improving generalization on the clean test set.
> >    - **Robust Generalization (Robust Accuracy):** Resolving gradient conflict allows the model to learn adversarially invariant features without interference from classification gradients, improving robust performance on adversarial examples.
> >    - **Harmonizing Clean and Robust Generalization:** Thus, resolving gradient conflict enables the model to learn shared representations that satisfy both clean and adversarial objectives, reducing the trade-off between clean and robust generalization.
> >
> > NOTE: Gradient conflict mitigation does not eliminate the fundamental trade-off. Instead, it shifts the trade-off curve, reducing the robustness-accuracy trade-off.
> >
> > **In conclusion, we addressed the robustness-accuracy trade-off from an optimization perspective.** While the theoretical relationship between AT with invariance regularization and the trade-off remains complex, **our results provide promising insights, demonstrating that existing strategies can be enhanced by improving training stability and convergence through careful loss function design.**
> >
> > We believe that a more formal discussion on this topic would definitely make a valuable contribution, which could be explored in a separate paper.
> >
> > ---
> > Thank you once again for your thoughtful questions and for dedicating significant time to reviewing our paper.
> > If you have any additional questions or suggestions, we would be happy to address them.

---

> > > ### Comment · Reviewer_zuTF · 2024-11-26
> > > **Thank the authors**
> > >
> > > Thank the authors. All my concerns are addressed. I have increased my score. :)

---

> > > > ### Author Response · Authors · 2024-11-26
> > > > **Response by Authors**
> > > >
> > > > We are grateful that the score has been increased and sincerely appreciate the significant time and effort you dedicated to reviewing our paper.
> > > >
> > > > We also deeply value your thoughtful questions on the theoretical aspects of our work, recognizing their importance. We will certainly keep them in mind for future research.
> > > >
> > > > Thank you so much once again!

---

### Official Review · Reviewer_JoTP · 2024-10-29

**Soundness:** 4
**Presentation:** 3
**Contribution:** 3
**Rating:** 8
**Confidence:** 4

**Summary:**

The authors propose AR-AT, which uses asymmetric invariance loss with stop-gradient operation and a predictor to avoid gradient conflict, and a split-BatchNorm (BN) structure to resolve the mixture distribution problem, to improve robustness-accuracy trade-off. The experiments demonstrate the effectiveness of this method.

**Strengths:**

1. The paper is easy-to-follow

2. The motivation of the proposed method is clear

**Weaknesses:**

1. In Figure 3, can you explain why $||z-z'||_2$ increases over time even though you add a regularization term? and why the use of stop-grad exacerbates this issue?

2. Line 200-201 & Figure 2: Plotting curves of only minimizing the second term can enhance your claim.

3. Table 17 indicates that solely resolving the mixture distribution problem can already improve the performance to a large extent. However, addressing gradient conflict alone does not contribute much to robustness, especially for (1)vs(2) and (5)vs(6). Thus, you should also report the standard deviations to make your results more convincing.

**Questions:**

1. Where is Figure 4?

2. What is the computational overhead of your method compared to other baselines?

---

> ### Author Response · Authors · 2024-11-19
> **Rebuttal by Authors**
>
> We sincerely appreciate your recognition that (1) the paper is easy to follow, (2) the motivation behind the proposed method is clear, and your positive assessment of our work.
>
> In the following, we address the raised concerns one-by-one.
>
> ---
> ## [W1: Why $||z' - z||_2$ increases over time even though you add a regularization term? and why the use of stop-grad exacerbates this issue?]
>
> We appreciate your detailed question and address each part below:
> - Q. Why does $||z' - z||_2$ increase over time?
> 	- This reflects the robustness-accuracy trade-off: as model accuracy improves, adversarial perturbations become easier to find, which in turn increases $||z' - z||_2$.
>    - To illustrate this further, we added Figure 13 in Appendix I.1. Figure 13 shows that this phenomenon also occurs in (1) adversarial training (AT) without regularization and (2) standard training, highlighting that this trade-off is common across training methods.
>
> - Q. Why does stop-grad exacerbate this issue?
>    - While stop-grad resolves gradient conflict and aids training convergence, it weakens invariance regularization by removing the second term in Eq. 5 ($(Dist(z', sg(z)) + Dist(z, sg(z'))) / 2$). This reduced regularization increases $||z' - z||_2$, intensifying the mixture distribution problem. Our split-BN strategy effectively mitigates this issue, ensuring the utility of stop-grad.
>
>  ---
> ## [W2: Line 200-201 & Figure 2: Plotting curves of only minimizing the second term can enhance your claim.]
>
> Thank you so much for your constructive suggestion.
>
> As suggested, we have plotted the curves of only minimizing the second term ($D(sg(z'), z)$) in Figure 2 (please refer to the updated paper).
> Interestingly, the results clearly show that this term is the primary source of gradient conflict with the classification loss. Specifically, minimizing this term led to **negative gradient similarity** and significantly larger proportion of parameters experiencing gradient conflict.
>
> We are so delighted with this insight and appreciate your high-quality feedback.
>
> ---
> ## [W3: Gradient conflict alone does not contribute much to robustness. Better to add standard deviation for Table 1.]
>
> As in lines 321-324 (highlighted in red in the updated paper), the fact that **gradient conflict alone does not contribute much to robustness is expected, and this is one of our key findings.**
> The issue arises from the exacerbated mixture distribution problem caused by stop-gradient, as shown in Figure 3. Table 1 clearly demonstrates that the benefit of stop-gradient becomes apparent when Split-BN is employed.
>
> Thank you for highlighting this. We will clarify this in the paper.
>
> ---
> ## [Q1: Where is Figure 4?]
> Thank you for pointing this out. We have corrected the error.
>
> ---
> ## [Q2: Computational overhead of your method compared to other baselines?]
>
> Thank you for your comment. In Appendix J, Table 19, we report the computational time of the baseline methods and ARAT.
>
> **ARAT is faster than TRADES and LBGAT.** This is because these methods use the $KL$ divergence ($KL(f_\theta(x)|| f_\theta(x'))$) for adversarial example generation, requiring forward passes for both clean and adversarial images. In contrast, ARAT uses the cross-entropy loss ($CE(f_\theta(x'), y)$), which only requires forwarding the adversarial images.
>
> Compared to AT, ARAT introduces a 15\% increase in time. This is due to (1) the need for forward passes of both clean and adversarial images for loss calculation (please refer to Table 10) and (2) additional updates for separate BatchNorm layers and the predictor MLP head.
>
> | Method | Clean | AA    |  Time (sec/batch)  | Time (rel. to AT) |
> |--------|-------|-------|:------------------:|-------------------|
> | AT     | 83.77 | 42.42 | 0.122 &plusmn; 0.045  | 1                 |
> | TRADES | 81.25 | 48.54 | 0.184 &plusmn; 0.112  | 1.51              |
> | LBGAT  | 85.00 | 48.85 | 0.183 &plusmn; 0.200  | 1.50              |
> | ARAT   | 87.82 | 49.02 | 0.140 &plusmn; 0.039  | 1.15              |
>
> ---
> Feel free to ask any further questions if anything is unclear. If you feel your concerns have been adequately addressed, we would appreciate it if you could adjust the score accordingly.

---

> > ### Comment · Reviewer_JoTP · 2024-11-20
> >
> > Thanks for your responses, most of my concerns have been addressed.
> >
> > However, I am still confused about why your method is faster than TRADES. As indicated in Table 10, AR-AT also needs to calculate the cross-entropy loss on clean samples (but with aux. BN), and you also mentioned that AR-AT needs forward passes of both clean and adversarial images for loss calculation. Thus, considering the aux. BN and predictor, your method should be slightly slower than TRADES when updating the parameters.
> >
> > What is the loss function you used when generating adversarial samples? Cross-entropy loss on adversarial samples? If so, your method could be faster than TRADES since TRADES generates adversarial samples based on KL divergence, which requires another forward pass of clean samples. Please clarify this.

---

> > > ### Author Response · Authors · 2024-11-20
> > > **Reply by Authors on Computational Cost**
> > >
> > > We are glad that most of your concerns have been addressed.
> > >
> > > **[Regarding Computational Cost of AR-AT]**
> > >
> > > Apologies for the earlier lack of clarity.
> > >
> > > Yes, you are correct:
> > > - AR-AT is faster than TRADES/LBGAT because it uses Cross-Entropy loss instead of KL-divergence for adversarial example generation, reducing the number of required forward passes by half.
> > > - Since adversarial example generation, which typically involves 10 iterations to update adversarial images, dominates the training time, the computational cost of updating the auxiliary BatchNorm and predictor parameters in AR-AT is negligible.
> > >
> > >
> > > We summarize the number of forward/backward passes below (denoted as "fwd","bw"):
> > > | Method   | Adv. Example Generation | Clean Loss | Adv. Loss | Inv. loss | Overall Loss BP | Total fwd | Total bp |
> > > |-------|------|-------|------|-------|-----|------|----|
> > > | TRADES   | $KL(f(x)\|\|f(x'))$, **10× (2 fwd + 1 bp)** | $CE(f(x), y)$, **1 fwd** | - | - | **1 bp** | **21 fwd** | **11 bp** |
> > > | LBGAT    | $MSE(f(x))\|\|g(x'))$, **10× (2 fwd + 1 bp)** | $CE(g(x), y)$), **1 fwd** | - | - | **1 bp** | **21 fwd**  | **11 bp** |
> > > | AR-AT    | $CE(f(x), y)$, **10× (1 fwd + 1 bp)** | $CE(f(x), y)$, **1 fwd** |  $CE(f(x'), y)$, **1 fwd** | (just extract features in fwd passes in clean/adv. loss) | **1 bp**| **12 fwd** | **11 bp** |
> > >
> > > The backward pass takes approximately 3x as long as the forward pass, which explains the time consumption of each method.
> > > We will clarify this in the paper.

---

> > > > ### Comment · Reviewer_JoTP · 2024-11-20
> > > >
> > > > Thanks for your elaboration. Additionally, I agree with Reviewer daZ8 that the comparison with more baselines is necessary. I suggest comparing your method with NuAT [1] and AdvLC [2], which also potent to improve robustness-accuracy tradeoff. I will adjust my score based on your updates.
> > > >
> > > > [1] Sriramanan et al. Towards efficient and effective adversarial training. NeurIPS 2021.
> > > >
> > > > [2] Li et al. Understanding and combating robust overfitting via input loss landscape analysis and regularization. Pattern Recognition 2023.

---

> > > > > ### Author Response · Authors · 2024-11-22
> > > > > **Response by Authors: Regarding More Baselines**
> > > > >
> > > > > Thank you for your valuable suggestion.
> > > > >
> > > > > Regarding more baselines, daZ8 suggested adding FAT[A], RandWeight[B], and HF[C], and JoTP recommended including NuAT[D] and AdvLC[E].
> > > > > To address these suggestions, we have created the following table for a comprehensive comparison.
> > > > >
> > > > > **NOTE:**
> > > > > - Suggested baseline papers report multiple method variants, with the best performance often achieved by method combinations. We compare both base and combined versions.
> > > > > - Inspired by this, we also evaluated **"ARAT+SWA"**, combining ARAT with Stochastic Weight Averaging (SWA)[F].
> > > > > - Combining ARAT with AWP [F] or HF [C] is a promising future direction, as these methods are orthogonal to ours, though they may require additional hyperparameter tuning.
> > > > >
> > > > > [Comparison of WRN-34-10 trained on CIFAR10 (sorted by Sum.)]
> > > > > | Method                      | Is comb.? | Clean | AA    | Sum.       |
> > > > > |-----------------------------|------------|-------|-------|------------|
> > > > > | AdvLC [E]                   |            | 82.23 | 45.11 | 127.34     |
> > > > > | AdvLC+SWA[E]                | &#x2714; | 82.09 | 49.73 | 131.82     |
> > > > > | AT                          |            | 86.06 | 46.26 | 132.32     |
> > > > > | FAT[A]                      |            | 89.34 | 43.05 | 132.39     |
> > > > > | NuAT2[D]                    |            | 84.76 | 51.27 | 136.03     |
> > > > > | TRADES                      |            | 84.33 | 51.75 | 136.08     |
> > > > > | TRADES+RandWeight[B]        |            | 85.51 | 54.00 | 139.51     |
> > > > > | TRADES+HF[C]                |            | 85.38 | 55.05 | 140.43     |
> > > > > | LBGAT                       |            | 88.19 | 52.56 | 140.75     |
> > > > > | NuAT2+WA[D]                 | &#x2714;       | 86.32 | 54.76 | 141.08     |
> > > > > | **(ours) ARAT**                 | | 90.89 | 50.77 | 141.66     |
> > > > > | AT+HF[C]                    |            | 87.53 | 55.58 | 143.11     |
> > > > > | TRADES+AWP+RandWeight[B]    | &#x2714;        | 86.10 | 57.10 | 143.20     |
> > > > > | **(ours) ARAT+SWA**             | &#x2714; | 90.17 | 54.45 | **144.62** |
> > > > >
> > > > >
> > > > > **Summary:**
> > > > > - **ARAT** outperforms the base methods **"FAT[A]"**, **"TRADES+RandWeight[B]"**, **"NuAT2[D]"**, and **"AdvLC [E]"**.
> > > > > - **AT+HF[C]** achieves better results than **ARAT**.
> > > > > - **ARAT** performs comparably to combined methods like **"TRADES+AWP+RandWeight[B]"** and **"NuAT2+WA[D]"**.
> > > > > - **"(ours) ARAT+SWA" achieves the best overall performance**, improving the robustness of AR-AT.
> > > > >
> > > > > We have included these results in Tables 3, 11, 12, and Figures 7 to 10. Thank you again for your constructive suggestions.
> > > > >
> > > > >
> > > > > ---
> > > > > **Nevertheless, we emphasize** that our main contribution lies in uncovering novel phenomena with thorough empirical evidence, and providing clear, step-by-step solutions. We believe this work offers valuable insights to the adversarial learning community.
> > > > >
> > > > > We are also pleased that the concerns raised by daZ8 and JoTP regarding our main claims and analysis have been addressed to their satisfaction.
> > > > >
> > > > > We humbly hope that reviewers will appreciate the thoroughness of our study as well as the novelty and originality of our work.
> > > > >
> > > > > ---
> > > > > **Reference.**
> > > > >
> > > > > (Baseline methods from daZ8:)
> > > > >
> > > > > [A] Attacks Which Do Not Kill Training Make Adversarial Learning Stronger. ICML 2020.
> > > > >
> > > > > [B] Randomized Adversarial Training via Taylor Expansion. CVPR 2023.
> > > > >
> > > > > [C] Focus on Hiders: Exploring Hidden Threats for Enhancing Adversarial Training. CVPR 2024.
> > > > >
> > > > > (Baseline methods from JoTP:)
> > > > >
> > > > > [D] Sriramanan et al. Towards efficient and effective adversarial training. NeurIPS 2021.
> > > > >
> > > > > [E] Li et al. Understanding and combating robust overfitting via input loss landscape analysis and regularization. Pattern Recognition 2023.
> > > > >
> > > > > (Other:)
> > > > >
> > > > > [F] Izmailov, Pavel, et al. "Averaging weights leads to wider optima and better generalization." arXiv preprint arXiv:1803.05407 (2018).
> > > > >
> > > > > [G] Wu, Dongxian, Shu-Tao Xia, and Yisen Wang. "Adversarial weight perturbation helps robust generalization." NeurIPS 2020.

---

> > > > > > ### Comment · Reviewer_JoTP · 2024-11-22
> > > > > >
> > > > > > Thanks for your updates. I have adjusted the score.

---

> > > > > > > ### Author Response · Authors · 2024-11-22
> > > > > > > **Response by Authors**
> > > > > > >
> > > > > > > We are delighted that the score has been updated to 'Acceptance.' We truly appreciate the valuable suggestions that have helped enhance the quality of our paper. Thank you once again for dedicating so much time and effort to this process.

---

### Official Review · Reviewer_Kdc6 · 2024-10-30

**Soundness:** 3
**Presentation:** 4
**Contribution:** 3
**Rating:** 8
**Confidence:** 3

**Summary:**

This paper proposed a bag of tricks to address two issues in siamese-based adversarial training, i.e., ``conflict gradients'' and the mixture distribution problem.

**Strengths:**

1.  The paper is well organized. The target issues are clear and each step of the solution is motivated and well represented.
2.  Experiments clearly show the effectiveness of each technique they proposed.

**Weaknesses:**

In Table 3, some baseline methods provide error bars, but others do not.

Moreover, considering the huge body of literature on adversarial training, I have concerns about whether the presented comparison with baseline methods is sufficient and comprehensive.

**Questions:**

1. Since the proposed method applies invariance regularization to multiple layers, I wonder if the baseline methods also adopt such multi-layer regularization.

2. Does your method introduce additional computation costs compared to baseline methods?

---

> ### Author Response · Authors · 2024-11-19
> **Rebuttal by Authors**
>
> We sincerely appreciate your recognition of (1) the clarity of the target issues and the motivation behind each solution, and (2) the clear presentation of experiments demonstrating the effectiveness of each technique.
>
> In the following, we address the raised concerns one-by-one.
>
> ---
> ## [W1: Regarding missing error bars for the baselines]
>
> Due to time constraints, we provided error bars only for our method and the state-of-the-art method, LBGAT. Still, we believe this adequately demonstrates the effectiveness of our approach.
> For ARREST, we were unable to run experiments since the authors did not release their code.
>
> ---
> ## [W2: Regarding more baseline methods]
>
> In the main paper, we focused on comparing with adversarial training (AT) methods that incorporate invariance regularization for the following reasons:
> - (1) Invariance regularization-based AT methods are the state-of-the-art approach to mitigate robustness-accuracy trade-off, as shown in Appendix B.
> - (1) We aimed to highlight that our contribution is understanding *the issues within the invariance regularization-based AT* and proposing solutions to improve performance.
>
> In Appendix B, we also provide comparisons between other AT methods, including methods not based on invariance regularization, demonstrating the strong performance of our proposed method, AR-AT.
>
> We believe this addresses your concern.
>
> ---
> ## [Q1: Invariance regularization types]
>
> TRADES, MART, and LBGAT apply invariance regularization in the logit space (the output of the classification layer, before softmax), while ARREST applies invariance regularization to the penultimate layer’s representation.
> This is summarized in Appendix A, Table 10.
>
> ---
> ## [Q2: Additional computation costs compared to baseline methods]
>
> Thank you for your comment. In Appendix J, Table 19, we report the computational time of the baseline methods and ARAT, trained on CIFAR-10 with ResNet-18 using a single NVIDIA A100 GPU and a batch size 128.
>
> **ARAT is faster than TRADES and LBGAT.** This is because these methods use the $KL$ divergence ($KL(f_\theta(x)|| f_\theta(x'))$) for adversarial example generation, requiring forward passes for both clean and adversarial images. In contrast, ARAT uses the cross-entropy loss ($CE(f_\theta(x'), y)$), which only requires forwarding the adversarial images.
>
> Compared to AT, ARAT introduces a 15\% increase in time. This is due to (1) the need for forward passes of both clean and adversarial images for loss calculation (please refer to Table 10) and (2) additional updates for separate BatchNorm layers and the predictor MLP head.
>
> | Method | Clean | AA    |  Time (sec/batch)  | Time (rel. to AT) |
> |--------|-------|-------|:------------------:|-------------------|
> | AT     | 83.77 | 42.42 | 0.122 &plusmn; 0.045  | 1                 |
> | TRADES | 81.25 | 48.54 | 0.184 &plusmn; 0.112  | 1.51              |
> | LBGAT  | 85.00 | 48.85 | 0.183 &plusmn; 0.200  | 1.50              |
> | ARAT   | 87.82 | 49.02 | 0.140 &plusmn; 0.039  | 1.15              |
>
> ---
> Feel free to ask any further questions if anything is unclear. If you feel your concerns have been adequately addressed, we would appreciate it if you could adjust the score accordingly.

---

> > ### Author Response · Authors · 2024-11-24
> > **Additional Comment by Authors**
> >
> > Thank you for your thorough review of our paper. We truly appreciate your insightful feedback and the time you invested in evaluating our work.
> >
> > We provide additional information to address your concerns:
> >
> > [W2] Regarding more baselines, we have added further comparisons in our response for JoTP, demonstrating ARAT's effectiveness. These results, which address JoTP's concern, are now included in Tables 3, 11, 12, and Figures 7 to 10 in the updated paper.
> >
> > [Q2] Regarding computational cost, we provided additional clarification in our response for JoTP, explaining why ARAT is faster than TRADES and LBGAT.
> >
> > We kindly request you to review our response to see if it sufficiently addresses your concerns, and reconsider the score accordingly.
> >
> > Thank you once again for your time and consideration.

---

> > > ### Comment · Reviewer_Kdc6 · 2024-11-24
> > >
> > > I appreciate the authors’ efforts in replying to my review.
> > > Most of my concerns have been thoroughly addressed, and I’m pleased to see that the additional experiments, such as those in Table 3, demonstrate even better performance. In my view, the empirical results comprehensively validate the proposed techniques' effectiveness. Furthermore, I do not think an empirical paper must necessarily include theoretical contributions. Based on this, I will increase my scores.

---

> > > > ### Author Response · Authors · 2024-11-24
> > > > **Response by Authors**
> > > >
> > > > We are grateful to hear that most of your concerns have been thoroughly addressed and that the score has been raised.
> > > > Your thoughtful review and dedicated efforts have significantly contributed to enhancing the quality of our paper. We sincerely appreciate the time you have taken in evaluating our work.
> > > >
> > > > With sincere thanks, Authors

---

### Official Review · Reviewer_daZ8 · 2024-11-01

**Soundness:** 3
**Presentation:** 3
**Contribution:** 2
**Rating:** 6
**Confidence:** 3

**Summary:**

This paper proposes a novel method called Asymmetric Representation-regularized Adversarial Training (AR-AT), aimed at improving the robustness-accuracy trade-off in adversarial training. AR-AT achieves this goal by addressing two key issues in invariance regularization: (1) the “gradient conflict” between invariance loss and classification objectives, and (2) the mixed distribution problem caused by the differences in input distributions between clean and adversarial samples. AR-AT introduces asymmetric invariance regularization, a stop-gradient operation, a predictor, and a split BatchNorm (BN) structure. Experimental results show that AR-AT outperforms existing methods across various settings and provides new insights into knowledge distillation-based defenses.

**Strengths:**

(1) This paper is well written and easy to follow.

(2) The paper validates the effectiveness of AR-AT through experiments on multiple datasets and model architectures, demonstrating superior performance in the robustness-accuracy trade-off.

(3) The paper not only presents a new method but also provides an in-depth analysis of the “gradient conflict” and mixed distribution problems in invariance regularization, offering new theoretical insights for the field of adversarial defense.

**Weaknesses:**

(1) The novelty of this paper maybe limited. The proposed stop-gradient operation and predictor are often adopted in self-supervised learning

(2) More evidence and experiments on mixed distribution problem should be claimed.

(3) There are a few minor errors, such as "clean representation z^' " in line 191.

**Questions:**

(1) The novelty of this paper may be somewhat limited, as the proposed stop-gradient operation and predictor are commonly utilized in self-supervised learning.

(2) Additional evidence and experiments addressing the mixed distribution problem should be provided.

(3) There are a few minor errors, such as the notation "clean representation Z^' " in line 191, which should be corrected.

(4) The ImageNet dataset should be incorporated into the experiments to enhance the robustness of the results.

---

> ### Author Response · Authors · 2024-11-19
> **Rebuttal by Authors**
>
> We sincerely appreciate recognizing (1) the clarity of our writing, (2) the strong empirical validation of AR-AT, and (3) our in-depth analysis of gradient conflict and mixture distribution problems, which provide novel and valuable theoretical insights for adversarial defense.
>
> Below, we address each of the raised concerns in detail.
>
> ---
> ## [W1, Q1] The novelty of the proposed stop-gradient operation and predictor
>
> We are glad to explain the novelty of our proposed method. We will clarify these points in the camera-ready version.
>
> ### **(Novelty 1) Distinct Motivation and Objective: Resolving "gradient conflict" by "Asymmetric structure"**
> While the loss functions may appear similar to SSL methods like SimSiam [A], our approach is tailored to address the **unique problem of "gradient conflict," distinct from the motivations in SSL.**
>
> Let us denote the distance function as $Dist(\cdot, \cdot)$, the stop-gradient operation as $sg(\cdot)$, and the predictor as $h(\cdot)$.
>
> - **[SSL setup]** In SSL, SimSiam uses two representations from two augmented images, $z1$ and $z2$, with the loss function:
> \begin{aligned}
>  L = (Dist(h(z1), sg(z2)) + Dist(h(z2), sg(z1))) / 2.
> \end{aligned}
> where the invariance regularization is **symmetric.**
>
> - **[AR-AT setup]** Let $z'$ represent the adversarial example and $z$ the clean image. In addition to classification losses, AR-AT (w/o predictor) employs an asymmetric invariance loss formulated as:
> \begin{aligned}
>  L_{inv} = Dist(z', sg(z)),
> \end{aligned}
> where the invariance regularization is **asymmetric.**
>
> - **[Key Difference]**  As discussed in Section 3.1, **we hypothesized that including the second term $Dist(sg(z'), z)$ (in Eq. 5) could lead to "corruption" of representations, causing "gradient conflict" between classification loss**. Figure 2 demonstrates that $Dist(sg(z'), z)$ induces gradient conflict (please refer to the updated paper).
>
> Thus, we emphasize that our approach goes beyond the simple application of SSL techniques: **Our asymmetric invariance loss is based on novel findings and distinct motivations, separate from SSL**. This makes our approach **fundamentally different from state-of-the-art defenses** like LBGAT and ARREST, which rely on knowledge distillation.
>
> ### **(Novelty 2) Novel Insights for Adversarial Defense**
>
> Crucially, the novelty of our work lies more in **offering new theoretical insights for adversarial defense**, as recognized by the reviewers, rather than in the technical details themselves.
> We **identified and tackled two key problems** of "gradient conflict" and mixture distribution issues in invariance regularization, supported by **thorough analysis and strong empirical evidence.**
>
> ---
> ## [W2,Q2: More evidence and experiments on mixture distribution problem]
>
> Thank you for your constructive suggestion.
> In Appendix I of the updated paper, we analyze the stability of BatchNorm (BN) statistics with and without Split-BN, providing additional evidence that supports our findings on the mixture distribution problem.
>
> In Figure 14, we plot the variance of the running mean $\mu$ of the BN layers computed at each epoch.
> **With the naive invariance loss ($L_{V0}$), the variance of the $\mu$ remains high throughout training. In contrast, split-BN stabilizes the updates of BN statistics by separating clean and adversarial BN statistics.**
> Specifically, (1) the variance of adversarial BN statistics is comparable or lower than that of the shared BN of the naive model, and (2) the variance for clean BN statistics is consistently lower.
> These results confirm that the split-BN stabilizes the estimation of BN statistics, effectively addressing the mixture distribution problem.
>
> We are delighted to strengthen our claim and highly appreciate your valuable feedback.
>
> ---
> ## [W3,Q3: Minor typos]
> We appreciate pointing out this error. It has been corrected in the updated version.
>
> ---
> ## [Q4: ImageNet results]
>
> Thank you for your suggestion. We agree that conducting experiments on a large-scale dataset like ImageNet is important, however, adversarial training on such datasets is extremely computationally expensive. This issue is common in the adversarial defense research community: most of the baseline methods, including ARREST and LBGAT, also do not provide results on ImageNet.
>
> In fact, adversarial training on ImageNet even with a small step size of 3 requires approximately 6 days with 4 V100 GPUs (https://github.com/MadryLab/robustness/issues/76), while adversarial training (10-steps) on CIFAR-10 takes just 2 hours with a single GPU.
>
> We acknowledge the significance of this issue but addressing it falls beyond the scope of our current work.
>
> ---
> Feel free to ask any further questions if anything is unclear. If you feel your concerns have been adequately addressed, we would appreciate it if you could adjust the score accordingly.
>
> ---
> **Reference**
>
> [A] Chen, Xinlei, and Kaiming He. "Exploring simple siamese representation learning." CVPR 2021.

---

> > ### Comment · Reviewer_daZ8 · 2024-11-20
> > **Response**
> >
> > Thank you for your responses. However, I still have some concerns regarding the experiments. The results on CIFAR and Imagenette do not seem particularly convincing.
> >
> > Additionally, I observed that most of the baseline models used in the experiments are from prior to 2023. It would be beneficial to include more recent and advanced baselines [1, 2, 3] for a more comprehensive evaluation.
> >
> > [1] Attacks Which Do Not Kill Training Make Adversarial Learning Stronger
> > [2] Randomized Adversarial Training via Taylor Expansion
> > [3] Focus on Hiders: Exploring Hidden Threats for Enhancing Adversarial Training.

---

> > > ### Author Response · Authors · 2024-11-22
> > > **Response by Authors**
> > >
> > > Thank you for your response.
> > >
> > > We are delighted that most of your concerns regarding [W1] the novelty of our method, [W2] evidence on the mixture distribution problem, and [W3] minor typos have been addressed to your satisfaction.
> > >
> > > We further address your concerns below:
> > > > The results on CIFAR and Imagenette do not seem particularly convincing.
> > >
> > > It is worth noting that none of the suggested baseline methods [A, B, C] provide ImageNet results due to computational constraints. Therefore, we emphasize that the lack of ImageNet results is not a specific limitation of our approach but rather a common challenge in adversarial training methods.
> > >
> > > > Additionally, I observed that most of the baseline models used in the experiments are from prior to 2023. It would be beneficial to include more recent and advanced baselines [1, 2, 3] for a more comprehensive evaluation.
> > >
> > > Thank you for your valuable suggestion.
> > > Regarding more baselines, daZ8 suggested adding FAT[A], RandWeight[B], and HF[C], and JoTP recommended including NuAT[D] and AdvLC[E].
> > > To address these suggestions, we have created the following table for a comprehensive comparison.
> > >
> > > **NOTE:**
> > > - Suggested baseline papers report multiple method variants, with the best performance often achieved by method combinations. We compare both base and combined versions.
> > > - Inspired by this, we also evaluated **"ARAT+SWA"**, combining ARAT with Stochastic Weight Averaging (SWA)[F].
> > > - Combining ARAT with AWP [F] or HF [C] is a promising future direction, as these methods are orthogonal to ours, though they may require additional hyperparameter tuning.
> > >
> > > [Comparison of WRN-34-10 trained on CIFAR10 (sorted by Sum.)]
> > > | Method                      | Is comb.? | Clean | AA    | Sum.       |
> > > |-----------------------------|------------|-------|-------|------------|
> > > | AdvLC [E]                   |            | 82.23 | 45.11 | 127.34     |
> > > | AdvLC+SWA[E]                | &#x2714; | 82.09 | 49.73 | 131.82     |
> > > | AT                          |            | 86.06 | 46.26 | 132.32     |
> > > | FAT[A]                      |            | 89.34 | 43.05 | 132.39     |
> > > | NuAT2[D]                    |            | 84.76 | 51.27 | 136.03     |
> > > | TRADES                      |            | 84.33 | 51.75 | 136.08     |
> > > | TRADES+RandWeight[B]        |            | 85.51 | 54.00 | 139.51     |
> > > | TRADES+HF[C]                |            | 85.38 | 55.05 | 140.43     |
> > > | LBGAT                       |            | 88.19 | 52.56 | 140.75     |
> > > | NuAT2+WA[D]                 | &#x2714;       | 86.32 | 54.76 | 141.08     |
> > > | **(ours) ARAT**                 | | 90.89 | 50.77 | 141.66     |
> > > | AT+HF[C]                    |            | 87.53 | 55.58 | 143.11     |
> > > | TRADES+AWP+RandWeight[B]    | &#x2714;        | 86.10 | 57.10 | 143.20     |
> > > | **(ours) ARAT+SWA**             | &#x2714; | 90.17 | 54.45 | **144.62** |
> > >
> > >
> > > **Summary:**
> > > - **ARAT** outperforms the base methods **"FAT[A]"**, **"TRADES+RandWeight[B]"**, **"NuAT2[D]"**, and **"AdvLC [E]"**.
> > > - **AT+HF[C]** achieves better results than **ARAT**.
> > > - **ARAT** performs comparably to combined methods like **"TRADES+AWP+RandWeight[B]"** and **"NuAT2+WA[D]"**.
> > > - **"(ours) ARAT+SWA" achieves the best overall performance**, improving the robustness of AR-AT.
> > >
> > > We have included these results in the paper.  Thank you again for your constructive suggestions.
> > >
> > >
> > > ---
> > > **Nevertheless, we emphasize** that our main contribution lies in uncovering novel phenomena with thorough empirical evidence, and providing clear, step-by-step solutions. We believe this work offers valuable insights to the adversarial learning community.
> > >
> > > We are also pleased that the concerns raised by daZ8 and JoTP regarding our main claims and analysis have been addressed to their satisfaction.
> > >
> > > We humbly hope that reviewers will appreciate the thoroughness of our study as well as the novelty and originality of our work.
> > >
> > > ---
> > > **Reference.**
> > >
> > > (Baseline methods from daZ8:)
> > >
> > > [A] Attacks Which Do Not Kill Training Make Adversarial Learning Stronger. ICML 2020.
> > >
> > > [B] Randomized Adversarial Training via Taylor Expansion. CVPR 2023.
> > >
> > > [C] Focus on Hiders: Exploring Hidden Threats for Enhancing Adversarial Training. CVPR 2024.
> > >
> > > (Baseline methods from JoTP:)
> > >
> > > [D] Sriramanan et al. Towards efficient and effective adversarial training. NeurIPS 2021.
> > >
> > > [E] Li et al. Understanding and combating robust overfitting via input loss landscape analysis and regularization. Pattern Recognition 2023.
> > >
> > > (Other:)
> > >
> > > [F] Izmailov, Pavel, et al. "Averaging weights leads to wider optima and better generalization." arXiv preprint arXiv:1803.05407 (2018).
> > >
> > > [G] Wu, Dongxian, Shu-Tao Xia, and Yisen Wang. "Adversarial weight perturbation helps robust generalization." NeurIPS 2020.

---

> > > > ### Author Response · Authors · 2024-11-24
> > > > **Official Comments by Authors**
> > > >
> > > > Dear Reviewer daZ8,
> > > >
> > > > Thank you once again for your thorough review of our paper. We truly appreciate your insightful feedback and the time you invested in evaluating our work.
> > > >
> > > > We kindly request you to review our response to see if it sufficiently addresses your concerns. Your feedback is of great importance to us.
> > > >
> > > > With sincere thanks,
> > > > Authors

---

### Author Response · Authors · 2024-11-19
**General Response by Authors**

We thank the reviewers for their valuable feedback and insightful comments. We greatly appreciate the constructive suggestions and have addressed them point by point.

**The paper has been updated based on these suggestions, with revisions highlighted in blue.**

If anything remains unclear, feel free to ask further questions. If you believe your concerns have been adequately addressed, we would be grateful if you could consider adjusting the score accordingly.

---

### Meta-Review · Area_Chair_Vxrg · 2024-12-20

**Metareview:**

This paper tackles the long-standing challenge of balancing robustness and accuracy in adversarial training by introducing a method called ARAT (Asymmetric Representation-regularized Adversarial Training). The authors focus on two key problems: gradient conflicts between invariance and classification objectives, which can hinder training, and the mixture distribution problem that arises from differences between clean and adversarial inputs. Their solution combines an asymmetric invariance loss with a stop-gradient operation and a split BatchNorm structure. The results show clear improvements over existing methods, backed by solid experiments and insightful analysis.

The paper is well-written, and particularly the use of asymmetric designs in AT is new and insightful. Reviewers note its clarity and strong empirical results and each component of the proposed method is carefully evaluated. The authors provided detailed explanations during the rebuttal to address questions about novelty and baseline comparisons. This effort significantly strengthened the reviewers' confidence in the work.

That said, some areas for improvement remain. A deeper theoretical exploration of the link between gradient conflict resolution and the robustness-accuracy trade-off would have made the paper even stronger. The lack of large-scale experiments, like those on ImageNet, was another point of critique.

Overall, the paper stands out for its  novel insights and solutions to adversarial training. It’s a solid contribution to the field and well-deserving of acceptance.

**Additional Comments On Reviewer Discussion:**

The discussion period was constructive, with the authors engaging thoughtfully with all reviewer concerns. Reviewer daZ8 raised questions about the novelty of the stop-gradient operation and predictor, as well as the absence of more recent baselines. The authors clarified how their approach differs from self-supervised learning techniques and expanded the experiments to include recent baselines, such as NuAT and AdvLC. This effort was appreciated, and daZ8 subsequently raised their score.

Reviewer JoTP focused on computational overhead and the need for additional plots to strengthen the claims. The authors provided a detailed explanation of their computational cost analysis and added visualizations, which directly addressed these concerns and led to another score increase.

Reviewer zuTF requested a deeper theoretical exploration of gradient conflict and its connection to the robustness-accuracy trade-off. While the authors could not provide a formal theoretical framework within the rebuttal timeline, they offered intuitive explanations and additional experiments, which satisfied the reviewer and led to an increase in their score.

Overall, the authors did an excellent job of addressing the reviewers’ points and strengthening their paper during the rebuttal period. Their efforts to clarify and expand on their contributions ensured unanimous support for acceptance.

---

### Decision · Program_Chairs · 2025-01-22

Accept (Poster)